# Constrained Hierarchical Deep Reinforcement Learning with Differentiable Formal Specifications

## Abstract

Formal logic specifications are a useful tool to describe desired agent behavior and have been explored as a means to shape rewards in Deep Reinforcement Learning (DRL) over a variety of problems and domains. Prior reward-shaping work, however, has failed to consider the possibility of making these specifications differentiable, which would yield a more informative signal of the objective via the specification gradient. This paper examines precisely such an approach by exploring a Lagrangian method to constrain policy updates using differentiable temporal logic specifications capable of associating logic formulae with real-valued quantitative semantics. This constrained learning mechanism is then used in a hierarchical setting where a high-level specification-guided neural network path planner works with a low-level control policy to navigate through planned waypoints. The effectiveness of our approach is demonstrated over four robot configurations with five different types of Signal Temporal Logic (STL) specifications. Our demo videos are collected in https://sites.google.com/view/schrl.

## 1 Introduction

Specifying tasks with precise and expressive temporal logic formal specifications has a long history (Pnueli, 1977; Kloetzer & Belta, 2008; Wongpiromsarn et al., 2012; Chaudhuri et al., 2021), but integrating these techniques into modern learning-based systems has been limited by the non-differentiability of the formulas used to construct these specifications. In the context of Deep Reinforcement Learning (DRL), a line of recent work (Li et al., 2017; Hasanbeig et al., 2019b; Jothimurugan et al., 2019; Icarte et al., 2022) tries to circumvent this difficulty by turning Linear Temporal Logic (LTL) specifications into reward functions used to train control policies for the specified tasks. The quantitative semantics introduced yields real-valued information about the task that can then be used by reinforcement learning agents via policy gradient methods. However, the sample complexity of such policy gradient approaches limits the scalability of these algorithms, especially when it comes to extracting reward functions from complex specifications (Yang et al., 2022). Moreover, these techniques do not consider how to effectively leverage the differentiability of the quantitative semantics associated with these specifications to yield a more accurate gradient than the policy gradient estimated from the LTL reward and samples.

Interestingly, as we show in this paper, this differentiability property can indeed be leveraged to meaningfully constrain policy updates. Previous approaches (Schulman et al., 2015; 2017; Achiam et al., 2017) constrain policy updates using KL-divergence and safety surrogate functions. For example, Achiam et al. (2017); Schulman et al. (2017) use Lagrangian methods for this purpose. Based on the same Lagrangian methods, we consider how to constrain policy updates with differentiable formal specifications (Leung et al., 2020; 2022) equipped with rich quantified semantics, expressed in the language of Signal Temporal Logic (STL)(Maler & Nickovic, 2004). This semantics gives us the ability to specify various tasks with logic formulas and realize them within a hierarchical reinforcement learning framework.

Instead of burdening a single policy to satisfy formal specifications and achieve control tasks simultaneously (Li et al., 2017; Hasanbeig et al., 2019b; Jothimurugan et al., 2019), we choose to learn a hierarchical policy. Hierarchical policies have proven to be effective in DRL with complex tasks

(Sutton et al., 1999; Nachum et al., 2018; Jothimurugan et al., 2021; Icarte et al., 2022). In contrast to previous DRL techniques integrated with LTL (Jothimurugan et al., 2021; Icarte et al., 2022), however, we replace multiple low-level options (Sutton et al., 1999) with a single goal-conditioned policy (Schaul et al., 2015; Nachum et al., 2018). A high-level planning policy, constrained by a formal specification, provides a sequence of goals to guide a low-level control policy to satisfy this specification. Additionally, because we wish for a learned policy to satisfy tasks as fast as possible, the high-level policy and low-level policy are jointly trained for both the satisfaction rate and the number of steps required to complete the objective. Finally, we also show novel applications of a neural-ODE policy as a high-level policy and integrate neural network-based predicate functions as part of our specification framework.

Our contributions are as follows. **(1)** We propose a programmable hierarchical reinforcement learning framework constrained by differentiable STL specifications, which avoids the sample complexity challenges of previous reward shaping work, and scales to benchmarks with high dimensional environments. **(2)** We show that the joint training of the high-level and low-level policy in this hierarchical framework provides better performance than training these components individually. **(3)** We demonstrate that our framework can be easily extended with neural predicates for complex specifications, such as irregular geometric obstacles that would be difficult to specify using purely symbolic primitives.

## 2 BACKGROUND

**Goal-Conditioned DRL**   Schaul et al. (2015) shows that training a single policy conditioned by multiple goals using only one neural network is feasible and can act as a universal option model (Yao et al., 2014). Given a goal $g$ and an agent observation $o_t$, the action $a_t = \pi(o_t \mid g)$ is predicted by the goal-condition policy $\pi(o_t \mid g)$. Intuitively, $a_t$ leads an agent to get "closer" to the goal $g$. Iteratively calling policy $\pi(o_t \mid g)$ in a loop will finally lead an agent to reach goal $g$.

**Two-layer Hierarchical DRL**   Combining a high-level planning policy with a low-level control policy can often expand the range of problems solvable by DRL algorithms (Florensa et al., 2017; Shu et al., 2018). Given a high-level planning policy $\pi^h : G^n \to G$ mapping all historical goals to the next goal, a low-level policy $\pi^l : O, G \to A$ maps the observation conditioned by a goal to action space $A$. At time step $t$, supposing that the low-level policy is toward $i$-th goal, the action is computed with $a_t = \pi^l(o_t|g_i)$, where $g_i = \pi^h([g_0, \ldots, g_{i-1}])$. In this work, the low-level policy $\pi^l$ is also called the control policy, and the high-level policy is identical to the planning policy.

**Lagrangian Methods in Constrained DRL**   Lagrangian methods solve constrained maximization problems. For a real vector $\mathbf{x}$, consider the equality-constrained problem:

$$\max_{\mathbf{x}} f(\mathbf{x}) \quad \text{s.t. } h(\mathbf{x}) = 0.$$

This can be expressed as an unconstrained problem with the Lagrange multiplier $\lambda$. Let $\mathcal{L}(\mathbf{x}, \lambda) = f(\mathbf{x}) + \lambda h(\mathbf{x})$,

$$(\mathbf{x}^*, \lambda^*) = \arg\min_{\lambda} \max_{\mathbf{x}} \mathcal{L}(\mathbf{x}, \lambda),$$

which can be solved by iteratively updating the primal variable $\mathbf{x}$ and dual variable $\lambda$ with gradients (Stooke et al., 2020). The $\lambda$ here acts as "dynamic" penalty coefficients for updating on real vector $\mathbf{x}$. Lagrangian methods are widely used in the policy gradient update of many popular constrained DRL algorithms (Schulman et al., 2017; Achiam et al., 2017). Note that to compute the gradient through the constraint function $h(\mathbf{x})$, the function $h$ must be differentiable. Since we want to constrain training with formal specifications, we, therefore, introduce a differentiable formal specification language in the following sections.

**TLTL Syntax and Operator Semantics**   The syntax of TLTL contains both first-order logic operators $\wedge$ (and), $\neg$ (not), $\vee$ (or), $\Rightarrow$ (implies), etc., and temporal operators $\bigcirc$ (next), $\Diamond_{[a,b]}$ (eventually), $\square_{[a,b]}$ (globally), $\mathcal{U}_{[a,b]}$ (until). The initial time $a$ and end time $b$ "truncate" a path. For example, $\square_{[a,b]}$ qualifies property that globally holds during time $a$ and $b$. The syntax of TLTL is recursively defined via the following grammar:

$$\phi := \top \mid \bot \mid \mathcal{P} \mid \neg\phi \mid \phi \wedge \psi \mid \phi \vee \psi \mid \phi \Rightarrow \psi \mid \bigcirc\phi \mid \Diamond_{[a,b]}\phi \mid \square_{[a,b]}\phi \mid \phi\,\mathcal{U}_{[a,b]}\psi$$

Given a path $\tau$ and state space $S$, a TLTL formula $\phi(\tau) : S^n \to \mathbb{B}$ maps a path to a Boolean value. Discrete Boolean values, however, make computing a continuous gradient infeasible. More specifically, we need quantitative semantics to enable differentiable specifications; we adapt Signal Temporal Logic (STL) (Maler & Nickovic, 2004) that equips TLTL specifications with quantitative semantics for this purpose.

**Quantitative Semantics** The full definition of the quantitative semantics is provided in Appendix C. We introduce four of the semantic rules here relevant to describe backpropagation through specifications. $\rho(\tau, \phi)$ denotes a real-valued function evaluated on path $\tau$ and represents the quantitative semantics of specification $\phi$. In this logic, we require that a true predicate ($\top$) always has a positive value $k$, which we write as $\rho(\tau_{[a:b]}, \top) = k$. Every user-defined (or neural network predicate) $\mathcal{P}$ must also be accompanied by its quantitative interpretation $\rho(s, \mathcal{P})$ in the real-value domain; here, if the return value is greater than 0, it denotes that the predicate is satisfied. For example, a reaching-task predicate $\mathcal{P}_{reach}$ can be defined as $\rho(s, \mathcal{P}_{reach}) := -\|s - g\| + c$, where $s$ is a state, $g$ is a goal, and $c$ is a positive threshold; if state $s$ is close to goal $g$, $\rho(s, \mathcal{P}_{reach})$ will be positive. In Sec. 3.1, we further show that a neural network can be learned as a quantitative measurement of a predicate. For the and ($\wedge$) operator, $\rho(\tau_{[a:b]}, \phi \wedge \psi) = \min(\rho(\tau_{[a:b]}, \phi), \rho(\tau_{[a:b]}, \psi))$. The quantitative semantics for temporal operator $\Diamond_{[a,b]}$ (**Eventually**) is defined as $\rho(\tau_{[a:b]}, \Diamond_{[a,b]}\phi) = \max_{t \in [a,b]} \rho(\tau_{[t:b]}, \phi)$.

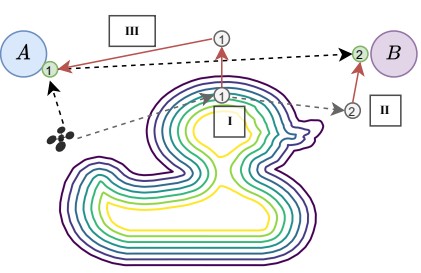

Figure 1: Demonstration of incorporating differentiable specifications and neural predicates to satisfy a *Coverage* specification that requires an agent to visit two goals ($A$ and $B$) while avoiding a complex obstacle (the "duck").

**Backpropagation through Specifications** The STL semantics defines a differentiable quantitative measurement for first-order logical and temporal operators, which allows gradients to backpropagate through them (Leung et al., 2020; 2022). Considering a coverage specification containing two goals $A, B$ with goal-reaching predicates $\phi_A, \phi_B$ as illustrated in Fig. 1, the specification is:

$$\phi_{cover} = \Diamond_{[0,T]}\phi_A \wedge \Diamond_{[0,T]}\phi_B \qquad (1)$$

The quantitative semantics of goal-reaching predicates $\phi_A$ and $\phi_B$, as well as $\wedge$ and $\Diamond_{[a,b]}$ are detailed in Sec. 2. We consider a path with three waypoints ($T = 2$). Given the initial path $\tau = [g_0, g_1, g_2]$, $\phi_{cover}$ can be evaluated as

$$\rho\left(\tau_{[0,2]}, \phi_{cover}\right) = \min\left(\max_{t \in [0,2]} \rho(\tau_{[t:2]}, \phi_A), \max_{t \in [0,2]} \rho(\tau_{[t:2]}, \phi_B)\right) \qquad (2)$$

All the operators in this semantics, including $\min, \max, \rho(\cdot, \phi.)$ and slicing ($[\cdot : \cdot]$), are differentiable and can be easily compiled to a computational graph with any auto-differentiation framework (e.g., PyTorch, JAX). Hence, we can differentiate through Eq. (2). Step **II** and **III** in Fig. 1 conceptually shows how backpropagation generates a path satisfying $\phi_{cover}$. A step-by-step example illustrating the entire updating process is provided in Appendix D.

## 3 APPROACH

### 3.1 NEURAL PREDICATES

The specification grammar allows specifications to use predicates whose interpretation is given by complex quantitative functions such as a neural network. For example, we can learn a neural network predicate for collision-avoidance tasks with neural Signed Distance Functions (SDF) (Park et al., 2019). A neural SDF $f_d(p, obj)$ represents the closest distance from a given point $p$ to the surface of the object $obj$. When $p$ is inside the object, the sign of $f_d$ is negative. The neural predicate $f_d$ can be seamlessly incorporated into STL specifications - when a waypoint is inside the object, $f_d$ is negative (evaluated as $\bot$); when a waypoint is outside the object, $f_d$ is evaluated as $\top$. For example, given a predicate of the duck obstacle shown in Fig. 1, $\phi_{duck} = f_d(p, duck)$, we can write a collision-avoidance specification $\Box_{[a,b]}\phi_{duck}$. Step **I** in Fig. 1 shows that increasing the value of

$\phi_{duck}$ with gradient ascent tweaks the waypoint to be outside of this duck obstacle. Details about the neural SDF are provided in Appendix E.

The predicate has been interpreted as a quantitative measurement of a given state (i.e., a function $\mathcal{P} : S \rightarrow \mathbb{R}$). The reward function is also exactly a function mapping from a state to a reward value. Thus, techniques such as inverse reinforcement learning Abbeel & Ng (2004); Ziebart et al. (2008) can be applied to learn a differentiable predicate without explicitly labeled data. Jha et al. (2019) provides a framework that makes the structure in STL also learnable. Our primary focus is not on investigating the diversity and application of neural predicates. We point out that our framework can be easily extended with differentiable functions generated using any relevant method, and leave details about the specific techniques that should be used in future work.

## 3.2 CONSTRAINED PATH PLANNING POLICY

Sec. 2 describes how gradients backpropagate through formal specification and how to use them to refine waypoints. However, directly tweaking the waypoint for different initial paths is computationally expensive for real-time deployment and cannot leverage learning techniques to yield better-quality plans. To alleviate these issues, we introduce a path-planning network.

**Path-Planning Network**  We employ a learnable recurrent neural-ODE planning neural network to generate planning paths as sequences of waypoints. Neural ODEs (Chen et al., 2018) can backpropagate with long paths effectively. The choice of a Recurrent Neural Network (RNN) is because temporal logic tasks are time-dependent. An example task may be to visit points $p_1, p_2, p_3$ sequentially. Without memorizing the visited points, a neural network cannot decide the next point that is to be reached. From a sampled initial position $g_0$, we run the neural ODE with a fixed timestep 1 (i.e., Euler ODE with timestep 1) and generate a goal path (i.e., waypoints) $[g_0, g_1, \ldots, g_T]$. Given a Gated Recurrent Unit cell (GRU) (Cho et al., 2014), and hidden state $h_i$,

$$g_{i+1} = g_i + \text{GRU}(g_i, h_i) \qquad (3)$$

The idea here is that the GRU cell predicts the change of a waypoint instead of directly predicting the next waypoint. An illustration figure for this structure is provided in the Appendix F (Fig 10b).

**Specification-Constrained Policy Gradient**  Because the GRU-ODE is suitable for path planning, we choose it as our high-level planning policy. For computing policy gradients, we modify the GRU-ODE as a stochastic policy. The Gaussian high-level policy is $\pi^h = \mathcal{N}(g_t, \sigma_t)$, where $g_t$ is the mean and $\sigma_t$ is the variance. $g_t$ is predicted by the GRU-ODE shown in Eq. (3), and $\sigma_t$ is predicted by a linear transform from the hidden state $h_t$. Consider the Lagrangian methods in Sec. 2 and objective function maximizing the expected reward under a constraint:

$$\mathcal{L}(\theta, \lambda) = \mathop{\mathbb{E}}_{\tau \sim \pi_\theta^h} \left( \pi_\theta^h(\tau) r^h(\tau) + \lambda(\epsilon - \rho(\tau, \phi)) \right), \qquad (4)$$

where $\epsilon > 0$ is a positive number. The introduction of $\epsilon$ is because we want $\rho(\tau, \phi)$ to be positive (evaluated as $\top$) when $\mathcal{L}$ converges optimally. While optimizing with gradient *ascent*, we first update the high-level policy parameters $\theta$ with the specification-constrained policy gradient:

$$\nabla \mathcal{L}(\theta, \lambda) = \mathop{\mathbb{E}}_{\tau \sim \pi_\theta^h} \left( \nabla_\theta \log \pi_\theta^h(\tau) r^h(\tau) - \lambda \nabla_\theta \rho(\tau, \phi) \right),$$

where $\nabla_\theta \log \pi^h(\tau) r^h(\tau)$ is the standard policy gradient, and $\rho(\tau, \phi)$ is a quantitative specification. We can compute $\nabla_\theta \rho(\tau, \phi)$ because $\tau$ is sampled from $\pi_h^\theta$. In practice, this requires that $\tau$ is sampled with reparameterization (Kingma & Welling, 2013). After updating the primal variable $\theta$, we update the dual variable $\lambda$ with gradient: $\nabla_\lambda \mathcal{L}(\theta, \lambda) = \epsilon - \rho(\tau, \phi)$ in a *descent* direction.

The reward $r^h(\tau)$ contains two parts: the quantitative specification $\rho(\tau, \phi)$ and the control reward $r^l(\tau)$ introduced later. We define $r^h(\tau) = \rho(\tau, \phi) + r^l(\tau)$. Note that we not only use $\rho(\tau, \phi)$ as part of the constraint function, but also consider it as part of reward $r^h$. A larger $\rho(\tau, \phi)$ means that the specification is "closer" to being satisfied. $r^l(\tau) = \sum_{i=0}^{T} r_t^l$ is a cumulative of control reward $r_t^l$ when following path $\tau$. A larger value for $r^l(\tau)$ means the generated path is easier to be followed by the low-level control policy. The involvement of $r^l(\tau)$ lets the planning policy also consider the capability of the low-level control policy.

### 3.3 CONTROL POLICY

The low-level control policy $\pi^l(o_t \mid \tilde{g})$ is a goal-conditioned policy, where $\tilde{g}$ is sampled from high-level policy $\pi^h$. $\pi^l(o_t \mid \tilde{g})$ is trained with PPO (Schulman et al., 2017), and the control reward is $r_t^l = \|\tilde{g} - x_{t-1}\|_2 - \|\tilde{g} - x_t\|_2$, where $x_t$ is the position of a robot at time $t$. $r_t^l$ will be positive if and only if a robot gets closer to the planned goal $\tilde{g}$ in one timestep. We train the control policy and planning policy jointly in a DRL loop. The detailed algorithm is in Appendix A.

## 4 EXPERIMENTS

We present experimental details related to robot dynamics, environments, and specifications in Sec. 4.1. The remaining subsections aim to answer the following questions: **Q1.** How does our approach compare to other DRL approaches with LTL specifications? **Q2.** What is the performance of the policies trained with our algorithm? **Q3.** What do we gain by jointly training the plan and control policy in a hierarchical structure? **Q4.** What if we replace the high-level policy with a path planner? **Q5.** Why do we need the GRU-ODE as the planning policy network? **Q6.** Why is the specification-constrained policy gradient better than the policy gradient with reward shaping?

### 4.1 DYNAMICS, ENVIRONMENTS, AND SPECIFICATIONS

**Dynamics and Environments**    All the robots we evaluated are shown in Fig 2. *Drone* is collected from PyBullet Drone (Panerati et al., 2021); *Point, Car* and *Doggo* are collected from safety gym (Achiam & Amodei, 2019). Their corresponding state and action space dimensions, denoted as (state dim, action dim), from (a)-(d) are (18, 18), (14,2), (26,2), and (58,12), resp.

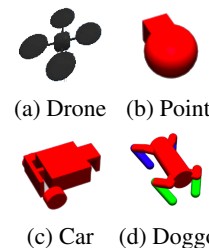

(a) Drone   (b) Point

(c) Car   (d) Doggo

Figure 2: four Robots

We provide the illustrative figures for the environments (i.e., plane, obstacles) in Appendix G. The *Drone* robot operates in an environment that contains the duck obstacle described earlier, while the other robots operate in an environment containing walls. The specification of the duck and walls are represented as neural SDF in our task specifications.

**Specifications**    We consider five types of specifications. They are *Sequence*, *Coverage*, *Branch*, *Loop*, and *Signal*. *Sequence* tasks require a robot to reach goals in a given order. In contrast, the *Coverage* tasks only require a robot to reach all goals without considering the order in which they are visited. *Branch* tasks are defined in terms of conditional behaviors. For example, if visiting $A$ then also visit $B$; if visiting $C$ then also visit $D$. *Loop* tasks ask robots to repeatedly visit a set of goals, while *Signal* tasks ask robots to repeatedly visit a set of goals until a certain condition is satisfied. All tasks also require robots to avoid obstacles modeled with neural network predicates. The formal definition of these five task types is provided in Appendix G.

### 4.2 TRAINING PERFORMANCE (**Q1**)

We compare the training performance of our algorithm, Specification Constrained Hierarchical Reinforcement Learning (SCHRL), with previous works, including CRM and DHRM from Icarte et al. (2022), and TLTL (Li et al., 2017). Approaches such as those described in Jothimurugan et al. (2019; 2021) cannot directly handle *Loop* and *Signal* tasks, so we cannot meaningfully compare their approach with ours in our experiments. HTLTL is a hierarchical version of TLTL, where a high-level planning policy is trained with a TLTL reward. The difference between this algorithm and ours is the objective function of high-level policy, where we remove the Lagrange item $\lambda(\epsilon - \rho(\tau, \phi))$ in Eq. (4). To the best of our knowledge, HTLTL has not been discussed in any previous work; it is introduced purely as an ablation experiment to assess the effectiveness of our algorithm.

CRM and TLTL are non-hierarchical algorithms. They need to control a robot and satisfy a task with a single controller. The state dimensions of our robots are at least 14 (on *Point*) and up to 58 (on *Doggo*); these continuous control tasks pose pressure on the control side, and CRM and TLTL fail to generate a reasonable controller as other algorithms with the same amount of samples on all

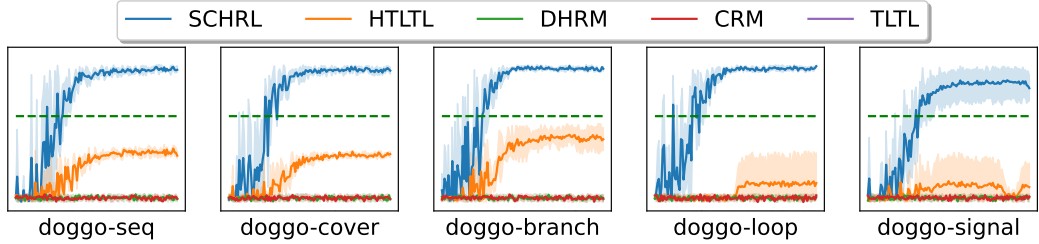

Figure 3: The quantitative specification scores (y-axis) against the simulation steps (x-axis) during training. We trained every algorithm with 5 different random seeds. The complete steps for *Doggo* are $1 \times 10^8$. Fig. 12 given in the Appendix provides the results of other robots. The tasks from left to right are *Sequence*, *Coverage*, *Branch*, *Loop*, and *Signal*, respectively. Any y-axis value above these green dash lines is greater than 0 and means a specification is satisfied (e.g., in a coverage task, the generated path gets closed enough to all the goals).

the tasks as shown in Fig. 3 and Fig. 12 in the Appendix H. DHRM is a hierarchical policy. However, it requires every transition in a reward machine to learn an option policy which is extremely challenging as specification and control complexity increase. It also cannot reuse the existing option policy for different transitions. For example, if two transitions in the reward machine have the same go-right requirement, DHRM will learn two separate policies. For *Doggo* with a more complex control task, the DHRM cannot learn a reasonable policy to achieve the tasks in given steps (see Fig. 3), but it can get higher scores than CRM and TLTL on *Sequence* and *Coverage* tasks on *Point* with simpler control tasks (see Fig. 12 in Appendix H).

The goal-conditioned policies applied in our HTLTL and SCHRL algorithms avoid building multiple policies. More importantly, on the control level, all sampled data is used to train a single goal-conditioned control policy instead of being distributed to different policies like DHRM. As a result, SCHRL and HTLTL perform better than DHRM given the same number of samples. The difference between SCHRL and HTLTL is in their use of specification constraints. The training results in Fig. 3 show that specification constraints significantly improve training both w.r.t the number of samples required and the best score it can achieve. Moreover, we noticed that HTLTL could reach higher rewards when compared with DHRM, CRM, and TLTL, which lends support to our use of a hierarchical RL structure with a goal-conditioned control policy and GRU-ODE planning policy.

## 4.3 SATISFYING RATE AND STEPS TO REACH (**Q2**)

The performance of trained policies is measured with two metrics. Firstly, we care about whether a robot can achieve a specified task. There may be many reasons that cause a robot to fail a task. For example, a robot can fall down, fail to avoid obstacles, or be misled by a badly planned path. Our policies demonstrate higher-than-$95\%$ satisfaction rates on all the tasks of the *Drone*, *Point*, *Car*, and *Doggo*, as shown in Table 1 and Table 3 in Appendix I.

We also care about how fast a robot can achieve a task. We summarize the number of steps needed to achieve a *Doggo* task in Table 1; Table 4 in Appendix I gives details for the other robots in our study. Note that "Loop" requires a robot to keep looping between some goals, so we report the number of steps when it cumulatively achieves the given goals ten times. The data in Table 1 is the average (before $\pm$) and standard deviation (after $\pm$) of five policies trained with different random seeds.

Table 1: Performance of Doggo Tasks

| Tasks | Seq | Cover | Branch | Loop | Signal |
|---|---|---|---|---|---|
| Satisfaction Rate | $0.98 \pm 0.02$ | $0.96 \pm 0.02$ | $0.97 \pm 0.01$ | $0.95 \pm 0.02$ | $0.97 \pm 0.01$ |
| Steps to Reach | $454.89 \pm 88.84$ | $545.95 \pm 129.76$ | $433.20 \pm 100.47$ | $1431.13 \pm 333.19$ | $1523.90 \pm 278.10$ |

The performance results in Table 1 are achieved by different components in our algorithm. Our ablation study in Sec. 4.4 shows that the absence of these components decreases performance.

## 4.4 ABLATION STUDIES

We conducted ablation studies on *Doggo* (the robot with the highest state/action dimensions) to justify our design choices. The data in Fig. 4 and Fig. 6 are summarized from 5 groups of 100 simulations. All the Steps to Reach are normalized within each task.

### 4.4.1 HIERARCHICAL POLICIES ABLATION (**Q3, Q4**)

We trained the planning and control policy jointly in each epoch, as detailed in Appendix A - when learning the goal-conditioned control policy, the control policy is specialized to the goals generated from the planning policy; when learning the planning policy, the control policy's reward is also considered in its reward function (as detailed in Sec. 3.2). To demonstrate the benefits of joint training, we replace the control policy with a general goal-conditioned control policy, and replace the planning policy with a gradient-based path planner, separately.

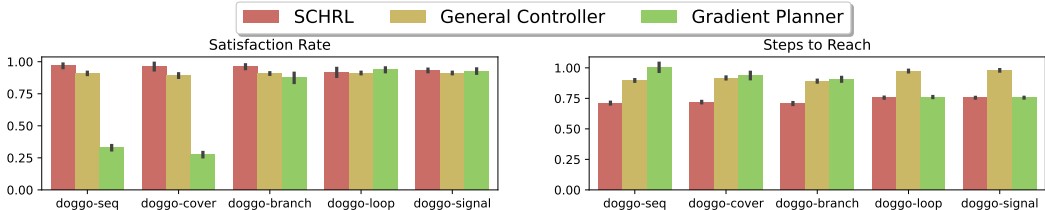

Figure 4: Policies Ablation

**General Goal-Conditioned Control Policy**   A general goal-conditioned policy can be obtained by training the policy with randomly sampled goals and the same reward function introduced in Sec. 3.3. The general goal-conditioned policy is used to replace the control policy in an SCHRL hierarchical policy; Fig. 4 shows comparative results. Although all general controllers (gold bars) have only a slightly lower satisfaction rate than SCHRL (red bar), they require a notably higher number of steps to reach. This is expected because general goal-conditioned policies do not adapt to the goals generated by a planning policy. For example, when a sharp turn happens in the planned path, a general goal-conditioned policy reacts more slowly than the policy learned by SCHRL. Note that on the *Signal* task, the difference in steps to reach can be as high as $24\%$.

**Gradient Planner**   A gradient planner can backpropagate through a differentiable specification, refine the waypoints with a gradient as shown in Fig 9, and generate a path. We replaced the planning policy with the gradient planner in an SCHRL hierarchical policy, and show the comparative results in Fig. 4. On task *Sequence*, *Coverage*, and *Branch*, the original SCHRL policy performs better in terms of both satisfaction rate and steps to reach. The largest difference in satisfaction rate ($66\%$) is on the *Coverage* task. On the *Sequence* task, the SCHRL policy requires $29\%$ fewer steps on average to achieve a task. On tasks *Loop* and *Signal*, the planning policy and the gradient planner perform similarly, and both have around $95\%$ satisfaction rate and similar steps to reach counts.

*Sequence*, *Coverage*, and *Branch* give flexibility to choose a waypoint (cyan circles in Fig 5a and Fig. 5b) between two goals. For example, a *Sequence* task may ask a robot to reach $A$ then $B$, but before reaching $B$, the robot can reach an additional waypoint in the middle. This middle waypoint can help a robot to circumvent an obstacle that intersects a straight-line path between two goals. However, as the specification only re-

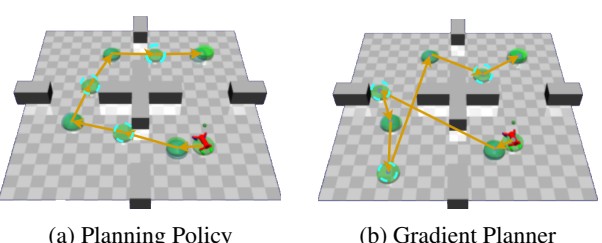

(a) Planning Policy          (b) Gradient Planner

Figure 5: Planning Paths

quires all the waypoints to be outside the region of an obstacle, a gradient planner only optimizing on the specification will not know how to place these middle waypoints appropriately. An example is shown in Fig. 5b where these middle waypoints are placed randomly. In contrast, because our

planning policy also considers the control reward as an optimization objective, a badly placed waypoint will result in the control policy failing to reach a goal and being assigned a bad reward. Hence, our planning policy can learn to place waypoints as shown in Fig. 5a. Because the path generated by the planning policy is shorter and yet does not intersect with obstacles, this path is easier to follow and can guide a robot to finish the task faster. In addition to the satisfaction rate and the steps to reach, we also note that a gradient planner needs additional computation time for different initialization positions, while a well-trained planning policy can plan faster with forward propagation. A comparsion on planning time is provided in Appendix J.

### 4.4.2 PLANNING NETWORK ABLATION (**Q5**)

Our planning policy is built upon a GRU-ODE, which has benefits both in terms of memory and gradient backpropagation. In Fig. 6, we show that GRU-ODE outperforms both Multilayer Perceptron (MLP) and GRU policy in almost every case. For example, on *Sequence* task, the GRU-ODE outperforms MLP and GRU $65\%$ and $51\%$ on the satisfaction rate and $29\%$ and $18\%$ on the steps to reach, respectively.

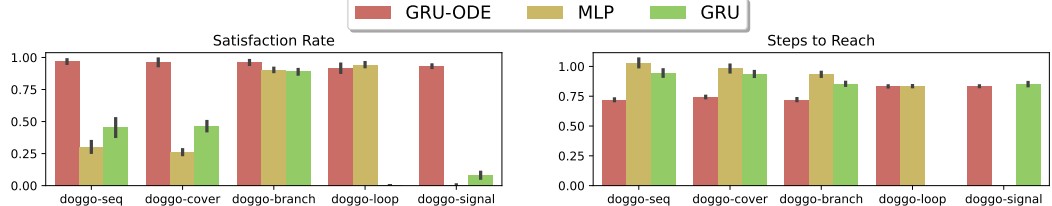

Figure 6: Planning Network Ablation. The GRU policy on *Loop* task and the MLP policy on *Signal* task failed to reach all the time, so their data is absent from the plot.

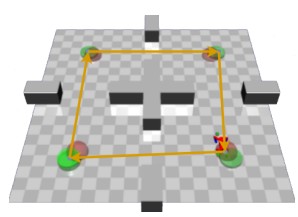

Figure 7: Loop task

One exception is that the MLP on the *Loop* task works equally well as GRU-ODE. This is because solving the *Loop* tasks does not require memory about the past. For example, in Fig. 7, to generate a path that satisfies the *Loop* constraint, when we know one goal is located at the upper-right green circle, a policy can always predict the next goal as the lower-right green circle to build one edge of a square loop path with whatever history it has. However, this is not the case for other tasks such as *Sequence*, where an agent needs to reach $A$ then reach $B$. If the robot reaches $B$ first, it still needs to go to $A$. Without a memory of the past, an MLP policy does not have the ability to make a decision in this case. This explains why it works badly on other tasks (e.g., $0\%$ satisfaction rate on *Signal* task). Although the GRU policy has a memory of the past, longer and more complex paths still pose training challenges. For example, the GRU policy has a low satisfaction rate on the *Loop* and *Signal* tasks, both of which have long paths. Details of these five tasks are provided in Appendix G.

### 4.4.3 CONSTRAINED POLICY GRADIENT ABLATION (**Q6**)

Table 2: PG vs CPG Ablation

| Tasks | PG | CPG |
|---|---|---|
| Seq | $9800 \pm 0$ (1 / 5) | **640** $\pm 102$ (**5** / 5) |
| Cover | N/A (0 / 5) | **780** $\pm 584$ (**5** / 5) |
| Branch | $9700 \pm 200$ (2 / 5) | **720** $\pm 527$ (**5** / 5) |
| Loop | N/A (0 / 5) | **1240** $\pm 150$ (**5** / 5) |
| Signal | N/A (0 / 5) | **1500** $\pm 420$ (**5** / 5) |

Previous reward-shaping work updates policy with sample-based Policy Gradient (PG). We run both the PG (eq. (4) minus $\lambda(\epsilon - \rho(\tau, \phi))$) and our Constrained Policy Gradient (CPG) (eq. (4)) 5 times on all the 5 high-level path planning tasks within 10000 gradient updates. The gradient updates will stop if over 95% of planned paths satisfy the STL specification. In each column, the number before and after $\pm$ is the mean and standard deviation of gradient updates, resp. N/A means in 5 runs, the algorithm failed to train a planning policy with over 95% satisfaction rate. The 2 numbers in the parentheses are the runs successfully trained by a policy and the total number of runs, resp. In all the 5 tasks evaluated, constrained policy gradient (with the Lagrangian term) results in 10x fewer gradient steps and can always successfully train a path planning policy.

## 5 RELATED WORK AND CONTRIBUTION

**Related Work** Integrating symbolic techniques with deep learning has led to recent interest in different learning domains (Manhaeve et al., 2018; Chaudhuri et al., 2021; Badreddine et al., 2022). LTL is a formal logic initially designed for system verification tasks (Pnueli, 1977). Because LTL can encode complex system behaviors in a precise manner, a line of work (Fainekos et al., 2005; Kloetzer & Belta, 2008; Kress-Gazit et al., 2009; Wongpiromsarn et al., 2012) considers its use in specifying various planning and control tasks. More recently, as advances in reinforcement learning have demonstrated its ability to solve challenging planning and control problems in an unknown, stochastic environment, a line of work has considered how to combine LTL-specified tasks with reinforcement learning. However, this approaches (Fu & Topcu, 2014; Li et al., 2017; Hasanbeig et al., 2018; 2019a;b; 2020; 2022; Jothimurugan et al., 2019; Jiang et al., 2020; Bozkurt et al., 2020; Xu et al., 2020; Icarte et al., 2022; Zhang & Kan, 2022) formulate this integration in terms of either continuous or discrete rewards. However, to the best of our knowledge, they have not considered applying STL (LTL equipped with quantified semantics) specifications in a constrained setting (Schulman et al., 2015; Achiam et al., 2017), which can be solved with the gradient-based Lagrangian methods (Stooke et al., 2020). Lagrangian methods require differentiable constraint functions. Although differentiable STL specifications are discussed in Leung et al. (2020; 2022), they do not apply differentiable STL in a DRL setting.

Jothimurugan et al. (2021); Icarte et al. (2022) notice that a hierarchical structure provides benefits when learning from LTL specifications. However, they require learning multiple policies as options (Sutton et al., 1999). Moreover, one policy in Jothimurugan et al. (2021); Icarte et al. (2022) is only used for one state transition on the automaton built from the specification, which makes such an approach problematic as specifications become more complex. In contrast, we select a different hierarchical structure based on a goal-conditioned control policy and a path planning policy (Levy et al., 2017; Nachum et al., 2018; Zhang et al., 2020); such an approach has been shown to be feasible theoretically (Schaul et al., 2015). Only one control policy is required in this setting, and all data samples can be used to train this single control policy.

Our work is also related to constrained reinforcement learning (Schulman et al., 2015; Achiam et al., 2017), programmable reinforcement learning (Andre & Russell, 2002; Yang et al., 2021; Qiu & Zhu, 2021), probabilistic temporal logic constraint (Hasanbeig et al., 2019b; Jansen et al., 2020), and gradient-based robot planning and control (Ratliff et al., 2009; Campana et al., 2016; Leung et al., 2020; Dawson & Fan, 2022). Unlike previous reinforcement learning algorithms that usually focus on safety constraints, we extend our constraints to be more expressive using formal logic. Programmable reinforcement learning also specifies the behaviors of a reinforcement learning agent in a programmable way (Andre & Russell, 2002; Yang et al., 2021); for example, Qiu & Zhu (2021) learns specifications with an idea inspired by neural architecture search. However, Andre & Russell (2002); Yang et al. (2021) also fall into the category of reward engineering with formal specifications. Qiu & Zhu (2021) design a domain-specific language without loops to automatically compose policies and thus cannot handle some of our tasks, such as *Loop*. Hasanbeig et al. (2019b); Jansen et al. (2020) synthesize controllers and shields with probabilistic guarantees for satisfying LTL specifications under uncertainty. The uncertainty is embedded in the environment (e.g., the uncertainty in the structure of the workspace, and the agent's actions). Such uncertainty also exists in our environments manifesting as stochastic dynamics and policies. Modeling these probabilistic behaviors and providing probabilistic guarantees of satisfying LTL specifications in hierarchical settings is an interesting direction for future work. Gradient-based motion planning (Ratliff et al., 2009; Campana et al., 2016; Leung et al., 2020; Dawson & Fan, 2022) also leverages backpropagation to generate plans. However, they typically only work when the dynamics (i.e., simulator) are known, differentiable and deterministic. Lastly, our high-level planning policy is related to planning networks (Qureshi et al., 2019), which supports fast online planning. However, learning such a planning neural network in a DRL loop, which benefits both the planning policy and control policy, has not been discussed in previous work.

**Conclusions** In this paper, we show how to leverage the differentiability of STL specifications to constrain policy updates in a hierarchical reinforcement learning framework, which leads to a framework co-optimizing the planning and control policies with challenging system dynamics and LTL tasks. We demonstrate that our approach outperforms other DRL techniques equipped with LTL specifications when the LTL specification is used only for reward shaping. We also justified the design choices of our approach with detailed ablation studies.

REPLICATION STATEMENT

Our code is available in https://github.com/a-n-onymous/schrl.git

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

## APPENDIX

## A    HIERARCHICAL LEARNING ALGORITHM

We summarize our learning algorithm in Algo. 1. Line 3 to 7 sample data with a control policy conditioned by the planned path $\tau$. Line 8 to 9 is the policy updates introduced in Sec. 3.2 and Sec. 3.3, respectively.

We point out that control and planning policies adapt to each other during training. The reward $r^h$ used for the planning policy contains the cumulative reward $r^l(\tau)$ of the control policy, enabling the planning policy to generate paths that are easier to follow. Meanwhile, the control policy is conditioned by the planning path, specializing it to the planning policy, allowing it to reach goals faster.

## B    TLTL SEMANTICS

The TLTL grammar rule is recursively defined as

$$\phi := \top \mid \bot \mid \mathcal{P} \mid \neg\phi \mid \phi \wedge \psi \mid \phi \vee \psi \mid \phi \Rightarrow \psi \mid \bigcirc\phi \mid \Diamond_{[a,b]}\phi \mid \Box_{[a,b]}\phi \mid \phi\,\mathcal{U}_{[a,b]}\psi$$

---

**Algorithm 1:** Specification-Constrained Hierarchical Reinforcement Learning

**Notions:** Spec $\phi$, Planning Policy $\pi^h$, Control Policy $\pi^l$, Rollout Buffer $\mathcal{B}$, Sequence $[\cdot]$, Observation $o$, Control Reward $r^l$, Planned Path $\tau$, Initial Position $g_0$.

```
1  B ← ∅;
2  for i = 1, 2, . . . , N₁ do
3      for j = 1, 2, . . . N₂ do
4          g₀ ← SampleInitial();
5          τ ← ForwardPath(πʰ, g₀) ;            // τ is waypoint sequence.
6          [o], [rˡ] = Rollout(πˡ, τ) ;          // Control policy follows τ.
7          B.add(τ, [o], [rˡ]);
8      UpdatePlanPolicy(πʰ, B) ;                 // Sec. 3.2
9      UpdateControlPolicy(πˡ, B) ;              // Sec. 3.3
10     B ← ∅;
```

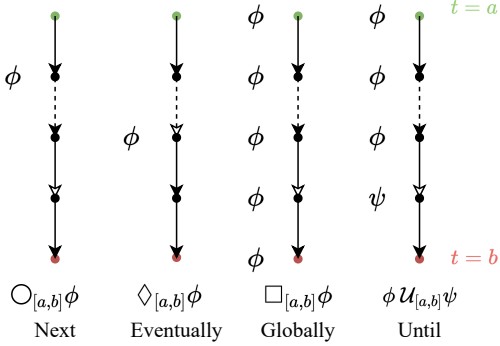

Figure 8: Temporal Operator Semantics.

The semantics of first-order logic operators ($\neg, \wedge, \vee, \Rightarrow$) is trivially defined. We summarize the semantics of temporal operators ($\bigcirc, \Diamond_{[a,b]}, \Box_{[a,b]}, \mathcal{U}_{[a,b]}$) in Fig. 8. All paths in Fig. 8 start at time $a$ and end at time $b$. **Next** ($\bigcirc\phi$) means that logical formula $\phi$ holds at next state in a path; **Eventually** ($\Diamond_{[a,b]}\phi$) says that $\phi$ will be satisfied at least once; **Globally** ($\Box_{[a,b]}\phi$) enforces the requirement that $\phi$ will always be satisfied; **Until** ($\phi\,\mathcal{U}_{[a,b]}\psi$) asserts that $\phi$ always holds until $\psi$ holds.

Additionally, we allow specifications to contain state-based predicates $\mathcal{P} : S \rightarrow \{\top, \bot\}$ that map a state to a Boolean value. For example, we can express a safe predicate $\mathcal{P}_{safe}(s) = \top \iff s \notin \mathcal{S}_u$ that holds precisely when $s$ does not belong to the set of unsafe states $\mathcal{S}_u$. We can combine these predicates freely in other specifications; for example, the following specification:

$$\left(\Box_{[a,b]}\mathcal{P}_{safe}\right) \wedge \left(\Diamond_{[a,b]}\mathcal{P}_{liveness}\right) \tag{5}$$

captures the requirement that an agent always stays in a safe region but eventually reaches a goal state ($\mathcal{P}_{liveness}$).

## C  Full Signal Temporal Logic Semantics

A specification's quantitative semantics (i.e., STL) maps a path to a real value. Its signature is: $\rho : S^n \rightarrow \mathbb{R}$. We provide the full semantics below:

$$\rho(\tau_{[a:b]}, \top) \qquad\qquad = k \qquad (k > 0)$$
$$\rho(\tau_{[a:b]}, \mathcal{P}) \qquad\qquad = \mathcal{F}(\tau_{[a]})$$
$$\rho(\tau_{[a:b]}, \neg\phi) \qquad\qquad = -\rho(\tau_{[a:b]}, \phi)$$
$$\rho(\tau_{[a:b]}, \phi \wedge \psi) \qquad\qquad = \min(\rho(\tau_{[a:b]}, \phi), \rho(\tau_{[a:b]}, \psi))$$
$$\rho(\tau_{[a:b]}, \phi \vee \psi) \qquad\qquad = \max(\rho(\tau_{[a:b]}, \phi), \rho(\tau_{[a:b]}, \psi))$$
$$\rho(\tau_{[a:b]}, \phi \Rightarrow \psi) \qquad\qquad = \max(\rho(\tau_{[a:b]}, \neg\phi), \rho(\tau_{[a:b]}, \psi))$$
$$\rho(\tau_{[a:b]}, \bigcirc\phi) \qquad\qquad = \rho(\tau_{[a+1:b]}, \phi)$$
$$\rho(\tau_{[a:b]}, \Box_{[a,b]}\phi) \qquad\qquad = \min_{t \in [a,b]} \rho(\tau_{[t:b]}, \phi)$$
$$\rho(\tau_{[a:b]}, \Diamond_{[a,b]}\phi) \qquad\qquad = \max_{t \in [a,b]} \rho(\tau_{[t:b]}, \phi)$$
$$\rho(\tau_{[a:b]}, \phi\, \mathcal{U}_{[a,b]}\psi) \qquad\qquad = \max_{t \in [a,b]} \left( \min \left( \rho(\tau_{[t:b]}, \psi), \min_{t' \in [a,t-1]} \rho(\tau_{[a:t']}, \phi) \right) \right)$$

$\top$ is defined as a positive value $k$. According to the definition of $\rho(\tau_{[a:b]}, \neg\phi)$, the $\bot$ is defined as $-k$. Value 0 is undefined in this semantics. For a predicate $\mathcal{P}$, we associate it with a differentiable function $\mathcal{F}(s) : S \to \mathbb{R}$. This function can be any differentiable function that maps a state to a real value. For example, one can define it as the negative distance to a goal plus a threshold or represent it with a neural network in Sec. 3.1. The other rules define how to compose different predicates in numerical ways. For example, the $\wedge$ is interpreted as a $\min$ operator between two predicates; the $\Box_{[a,b]}$ (globally) operator is interpreted as a $\min$ operator over a finite path. The semantics of $\mathcal{U}_{[a,b]}$ (until) is non-trivial. It comes from an equivalent formula of $\mathcal{U}_{[a,b]}$:

$$\phi\, \mathcal{U}_{[a,b]}\psi := \Diamond_{[a,b]}(\psi \wedge \Box_{[a,t-1]}\phi)$$

where $t$ is a time when $\psi$ holds.

All the operators in this semantics, including $\min, \max$, predicate function $\mathcal{F}(\cdot, \phi)$, index ($[\cdot]$), and slicing ($[\cdot : \cdot]$), are differentiable and can be easily compiled to a computational graph with any auto-differentiation framework (e.g., PyTorch, JAX).

The min and max can in the semantics can be soften as $\widetilde{min}$ and $\widetilde{max}$:

$$\widetilde{min} = softmin(v) \cdot v^T$$
$$\widetilde{max} = softmax(v) \cdot v^T$$

where

$$softmin(v) = \frac{e^{-v}}{\sum_i e^{-v_i}}$$
$$softmax(v) = \frac{e^{v}}{\sum_i e^{v_i}}$$

Jha et al. (2019); Leung et al. (2020; 2022); Gilpin et al. (2021) demonstrated that such soften thick benifits gradient-based STL path planning. Experiments about using soft STL (with soft min and max) and hard STL (with general min and max) as Lagrangian terms is provided in the Appendix. L

## D    BACKPROPAGATE THROUGH A COVERAGE SPECIFICATION

We discuss how the gradient backpropagates a coverage specification example with two goals $A, B$ specified by $\phi_A, \phi_B$, resp:

$$\phi_{cover} = \Diamond_{[0,T]}\phi_A \wedge \Diamond_{[0,T]}\phi_B$$

The goal-reaching predicates $\phi_A$ and $\phi_B$ are defined as $\rho(s, \phi_{\cdot}) := -\|s - g\| + c$, where $s$ is a state, $g$ is the position of either $A$ or $B$, and $c$ is a positive threshold. We consider a path with three waypoints ($T = 2$). $\phi_{cover}$ can be evaluated as

$$\rho\left(\tau_{[0,2]}, \phi_{cover}\right) = \min\left(\max_{t \in [0,2]} \rho(\tau_{[t:2]}, \phi_A), \max_{t \in [0,2]} \rho(\tau_{[t:2]}, \phi_B)\right).$$

The yellow dot in Fig. 9 is the initial waypoint and can be any valid initial state. The other two green points are planning waypoints. Because we want the path to satisfy the specification, we maximize the $\rho\left(\tau_{[0,2]}, \phi_{cover}\right)$ using its gradient. Fig. 9 shows how the gradient backpropagates through the Eq. (2). Before presenting the details, we point out that the gradient of $\min$ and $\max$ is only a gradient masking operator. For example, given $x_1 < x_2$, $\min(x_1, x_2)$ will not backpropagate any gradient to the variable $x_2$.

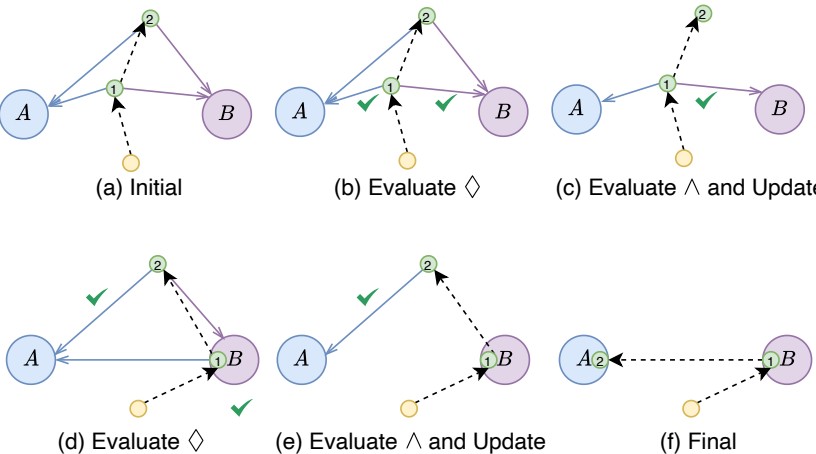

Figure 9: Gradient backpropagation through a coverage specification. (a) shows the initial path and the distance to $A$ and $B$. We do not modify the initial waypoint. It is detached from the gradient computation. (b) $\Diamond_{[0,2]}$ (eventually) is a $\max$ operator over a path. It will select the two short distances (larger negative distance, marked with check logos) to backpropagate. (c) $\wedge$ is evaluated as a $\min$ operator. Thus, it selects the longer distance (smaller negative distance, marked with a check logo) to backpropagate. Now, we tweak waypoint 1 to $B$ with gradient. This figure simplifies the gradient updating process to one step for illustration purposes. In practice, the update can be more tortuous, and waypoint 1 may take several steps to reach $B$. (d) and (e) are similar to (b) and (c), respectively, but they update on waypoint 2. (f) shows the final planned path. This path covers both $A$ and $B$.

# E    DETAILS ABOUT NEURAL SDF

The neural SDF fits an SDF $f_d(p, obj)$ with a neural network. A neural SDF $f_d(p, obj)$ computes the closest distance from a given point $p$ to the surface of object $obj$. When $p$ is inside the object, the sign of $f_d$ is negative. One can easily get an SDF with an object mesh (Park et al., 2019). However, the SDF gotten in this way is usually non-differentiable. Instead, we sample a dataset of position-distance pairs $(p, d)$ with such an SDF and train a differentiable neural network to predicate the closest distance $d$ with a position $p$.

Fig. 10a is the contour plot of a neural SDF. This neural SDF is built from the duck obstacle in (e)-(f) in Fig. 11. More precisely, it is the contour plot for a cross-section of this 3D duck when $y = 0$. The purple outline represents value 0, with values inward decreasing monotonically. This neural SDF can naturally represent the differentiable function $\mathcal{F}$ of the STL semantics in Appendix C. For example, when we want to define a collision-avoidance predicate $\phi_{safe}$ on this duck obstacle, if a waypoint is inside this obstacle, the $\phi_{safe}$ should be interpreted as a negative value under STL semantics (i.e., $\bot$). Similarly, an outside waypoint should have a positive value (i.e., $\top$). Moreover,

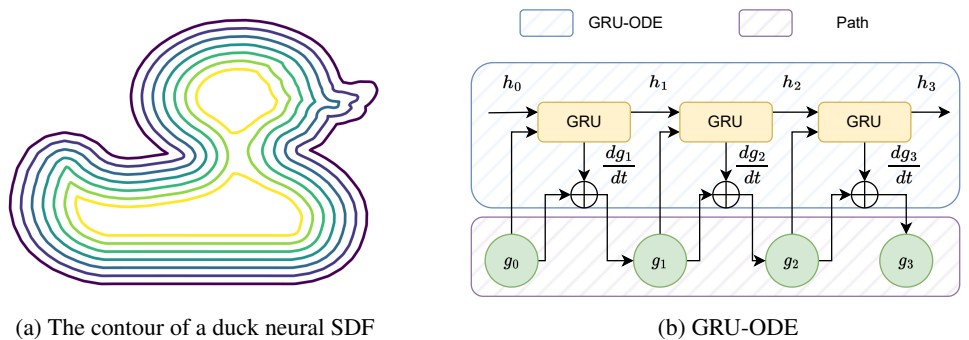

(a) The contour of a duck neural SDF      (b) GRU-ODE

Figure 10: (a) The purple outline represents value 0 with values inward decreasing monotonically. (b) The initial point $g_0$ is given. Feeding it to the GRU cell generates the time derivative $\frac{dg_0}{gt}$; $g_1$ is generated by adding the $\frac{dg_0}{dt}$ and $g_0$. Iterating this procedure generate a path $\tau = [g_0, g_1, g_2, g_3]$.

when maximizing an STL formula with gradients, the gradients will "push" a waypoint inside the obstacle to the outside, as shown in Fig. 1.

## F  DETAILS ABOUT GRU-ODE

We consider a neural ODE with a fixed timestep 1 (i.e., Euler ODE with timestep 1). Given a GRU cell, goal $g_i$ and hidden state $h_i$, the next goal $g_{i+1} = g_i + \text{GRU}(g_i, h_i)$. Fig. 10b illustrates the forward computation of the GRU-ODE. Note that

$$g_i = g_0 + \text{GRU}(g_0, h_0) + \cdots + \text{GRU}(g_{i-1}, h_{i-1}),$$

making it easy to backpropagate from $g_i$ to any previous forward of GRU. This idea is similar to the ResNet (He et al., 2016), which alleviates gradient vanishing in a long-term backpropagation.

## G  TASKS AND SPECIFICATIONS

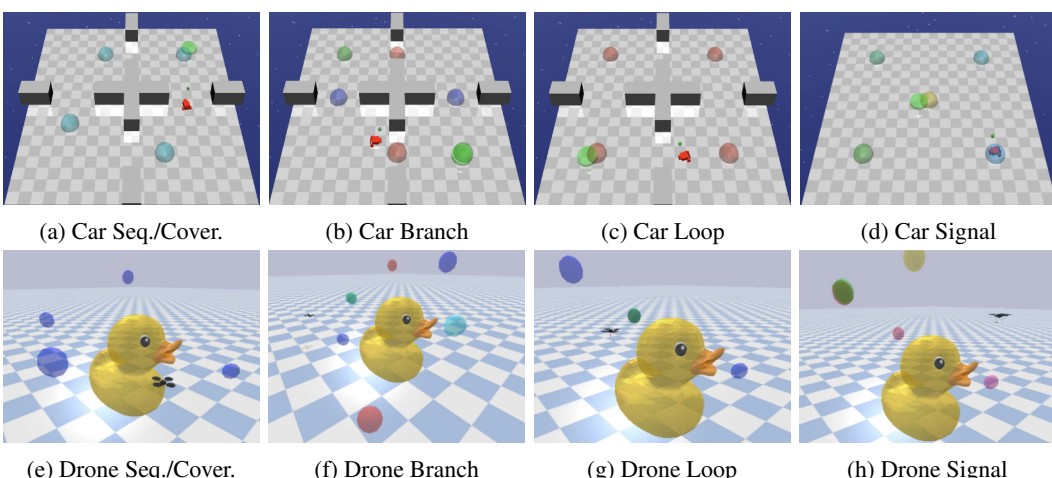

(a) Car Seq./Cover.    (b) Car Branch    (c) Car Loop    (d) Car Signal

(e) Drone Seq./Cover.    (f) Drone Branch    (g) Drone Loop    (h) Drone Signal

Figure 11: Illustration for All Tasks. The *Point* and *Doggo* have the same setting with *Car* on all the tasks. The robots (*Point*, *Car* and *Doggo*) in Fig. 11b (*Branch*) are randomly initialized in the bottom-left room; all the other tasks in the 4-room environment randomly initialize robots (*Point*, *Car* and *Doggo*) in the bottom-right room. All the *Drone* tasks randomly initialize the *Drone* in a box space with lower-bound $(-5, 4, 3)$ and upper-bound $(-4, 5, 8)$.

The *Point* and *Doggo* have the same setting with *Car* on all the tasks.

The *Sequential* task requires a robot to visit a sequence of goals in order.

$$\bigwedge_{i=0}^{N} \Diamond_{[t_i, t_{i+1}-1]} \phi_i, \tag{6}$$

where $t_i < t_{i+1}$ and $\phi_i$ is a goal spec. This formula means eventually reaching all the goals in specified time slots.

In Fig. 11a, a car is required to visit the upper-right ($\phi_0$), upper-left($\phi_1$), bottom-left($\phi_2$), and bottom-right($\phi_3$) blue spheres sequentially. The time slot assigned to $\phi_0$, $\phi_1$, $\phi_2$, and $\phi_3$ are $[1, 2]$, $[3, 4]$, $[5, 6]$, and $[7, 8]$, respectively. The time here represents the index of a waypoint in a path. In other words, in a path with a length 8, the first or the second waypoint in this path should reach the upper-right blue spheres. Similarly, the *Drone Sequence* also requires the robot to visit 4 blue balls in Fig. 11e in the slots $[1, 2]$, $[3, 4]$, $[5, 6]$, and $[7, 8]$.

As introduced in Sec. 2, the *Coverage* task's formula is

$$\bigwedge_{i=0}^{N} \Diamond_{[1,T]} \phi_i. \tag{7}$$

The *Coverage* task shares the same map with the *Sequence* task. For *Point*, *Car*, and *Doggo*, they use the same map in Fig. 11a, and *Drone* uses the map in Fig 11e. However, in the *Coverage* task, a robot is only asked to visit the four spheres, but discards the order. In Eq. 7, the $T$ is assigned as 8 for all the robots.

*Branch* task can be seen as a collection of $if \dots then \dots$ clusters, e.g., if visiting $A$ then visiting $B$, if visiting $C$ then visiting $D$. We encode this type of task as

$$\bigvee_{i=0}^{\frac{N}{2}} \left( \Diamond_{[1,T]} \phi_i \wedge \Box_{[1,T]} \left( \phi_i \Rightarrow \Diamond_{[t,T]} \phi_{2i} \right) \right), \tag{8}$$

where $N$ is even. This formula says eventually satisfying $\phi_i$, and globally $\phi_i$ implies that $\phi_{2i}$ will be satisfied eventually. The $\phi_i$ and $\phi_{2i}$ has the same color in Fig. 11b and Fig. 11f. This specification defines that a robot must visit one goal, and it will visit the other one with the same color eventually. For all the robots, $t = 2$, $T = 3$ and $N = 6$ in Eq.(8). All the *Point*, *Car*, and *Doggo* use the same map in Fig. 11b

The *Loop* task asks a robot to loop among some waypoints. It can be written as

$$\Box_{[1,T]} \left( \bigvee_{i=0}^{N} \phi_i \wedge \bigwedge_{i=0}^{N} \left( \phi_i \Rightarrow \neg \bigcirc \phi_i \right) \right). \tag{9}$$

This formula says that globally, any of $\phi_i$ should be satisfied, and $\phi_i$ will not be satisfied next time. A waypoint in a path will also stay in one of the giving goals specified by $\phi_i$, and the next waypoint will not stay here for the next time. In Fig. 11c the robot will loop among the four red spheres, and the *Drone* will loop among the three blue balls in Fig. 11g. The TLTL only supports finite paths, so we have to set a terminated time ($T$ in Eq. (9)). For all the robots, $T = 20$.

*Signal* task is a loop task with until operators:

$$\Diamond_{[1,T]} \phi_{N+2} \wedge \left( \bigvee_{i=0}^{N} \phi_i \wedge \bigwedge_{i=0}^{N} \left( \phi_i \Rightarrow \neg \bigcirc \phi_i \right) \right) \mathcal{U}_{[1,T]} \phi_{N+1}. \tag{10}$$

This means looping until $\phi_{N+1}$ is satisfied and eventually visits the position specified by $\phi_{N+2}$. For all tasks, the $\phi_{N+1}$ is a counter predicate; it is evaluated as $-1(\bot)$ until the 10th waypoint in a path, which means a robot needs to visit the position specified by $\phi_0, \dots, \phi_N$ 10 times, and eventually visits the position specified by $\phi_{N+2}$. In our experiments, $T = 11$ for all the tasks. In Fig. 11d, a robot will loop among the four corner spheres and visit these spheres ten times, and eventually visit the center sphere. In Fig. 11h, the *Drone* will keep visiting three pink balls ten times and eventually visit the highest yellow one.

All these formulas will be appended with an obstacle-avoidance cluster $\wedge \Box_{[1,T]} \phi_{obs}$. For the *Drone* tasks, the $\phi_{obs}$ is a duck neural SDF introduced in Sec 3.1. For the *Point*, *Car*, and *Doggo* tasks, the $\phi_{obs}$ is a neural SDF of all the walls.

## H    MORE TRAINING CURVES

We provide the training curves for *Drone*, *Point*, and *Car* in this section. All the data is collected with training with five different random seeds.

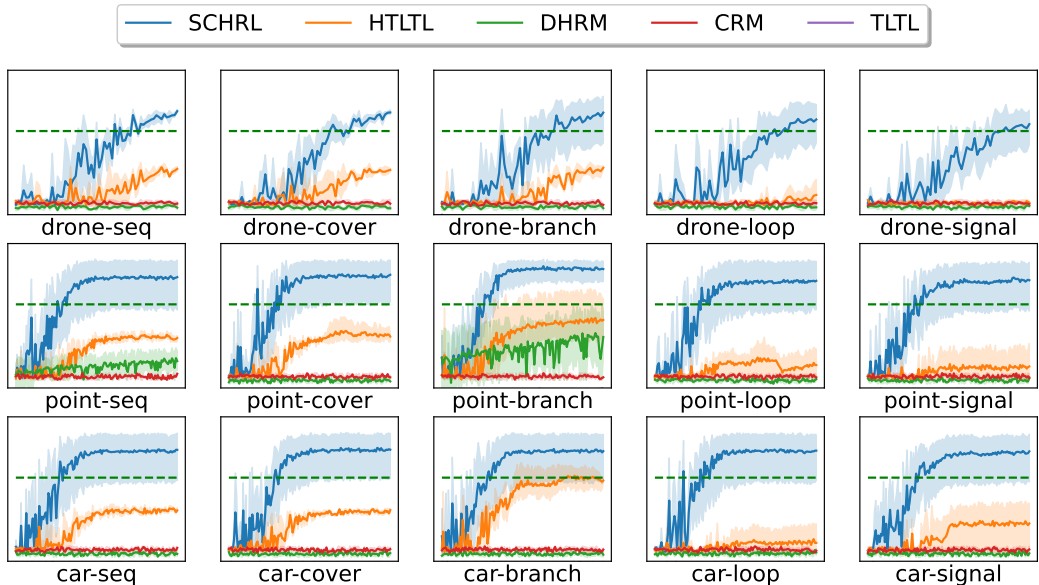

Figure 12: The quantitative specification scores (y-axis) against the simulation steps (x-axis) during training.

The SCHRL outperforms all the other algorithms for all robot dynamics and tasks. In the given steps, the DHRM learns better policies than non-hierarchical CRM and LTL on the *Sequence* and *Coverage* tasks on *Point*.

## I    ADDITIONAL POLICY PERFORMANCE

We report the SCHRL policy performance of *Drone*, *Point*, and *Car* in this section. The data is collected with 5 groups of 100 times simulations.

Table 3: Satisfaction Rates of *Drone*, *Point*, and *Car*

| Tasks | Seq | Cover | Branch | Loop | Signal |
|-------|-----|-------|--------|------|--------|
| Drone | 0.96 ± 0.01 | 1.00 ± 0.00 | 0.97 ± 0.01 | 1.00 ± 0.00 | 1.00 ± 0.00 |
| Point | 1.00 ± 0.00 | 1.00 ± 0.00 | 1.00 ± 0.00 | 1.00 ± 0.00 | 0.99 ± 0.01 |
| Car | 0.98 ± 0.00 | 0.96 ± 0.01 | 0.95 ± 0.01 | 0.94 ± 0.01 | 0.95 ± 0.01 |

The trained SCHRL policies generated decent satisfaction rates on all the dynamics and tasks. The worst case appeared in the *Loop* task of *Car*, where it has a $95\%$ satisfaction rate on average. The *Point* robot almost can always satisfy a task expecting the *Signal* task with $99\%$ satisfaction rate on average.

Table 4: Achieving Steps of *Drone*, *Point*, and *Car*

| | Seq | Cover | Branch | Loop | Signal |
|-------|-----|-------|--------|------|--------|
| Drone | 1453.36 ± 373.71 | 1165.97 ± 131.42 | 761.66 ± 153.76 | 2342.76 ± 213.68 | 2859.54 ± 220.96 |
| Point | 601.93 ± 18.70 | 669.93 ± 17.51 | 546.96 ± 34.00 | 1861.21 ± 126.20 | 1912.04 ± 174.05 |
| Car | 547.47 ± 97.18 | 592.56 ± 153.10 | 508.48 ± 136.58 | 1611.68 ± 387.99 | 1802.66 ± 26.79 |

The steps to achieve tasks are summarized in Table 4. All the *Loop* and *Signal* tasks have longer planned paths. They are with higher achieving steps than the *Sequence*, *Cover*, and *Branch* for each robot.

## J  Planning Time

One benefit we mentioned in Sec. 4.4.1 is planning time. Compared with Leung et al. (2020; 2022); Gilpin et al. (2021), our approach requires less planning time when deployed online.

Table 5: Planning Time (in sec.) Comparision

| Tasks | Planning Net | MIP | Gradient |
|---|---|---|---|
| Seq | **0.001722** | 0.096137 | 0.842844 |
| Cover | **0.001770** | 0.218563 | 3.484746 |
| Branch | **0.001437** | 0.078942 | 7.204325 |
| Loop | **0.004844** | 7.423811 | 10.480951 |
| Signal | **0.002383** | 7.853121 | 2.670637 |

The planning policy is a neural network that generates planning paths from randomly sampled initial states. A forward propagation in milliseconds will produce a path with MIP (Gilpin et al., 2021) or gradient techniques (Leung et al., 2020; 2022) for several seconds to solve. We quantify the planning time in Table 5.

The time in Table 5 is the average time (in sec.) over 100 runs. Our planning policy is at least 55X faster than MIP planners and 489X faster than gradient-based planners.

## K  Goal-Conditioned Policy and Options

Previous work learns one control policy (option) for each transition in the automaton built with an LTL specification. For example, a sequence task with four goals will require four policies.

Table 6: Samples Required until Convergence for Point

| | Goal-Conditioned Policy | 1 Option | 2 Options | 4 Options |
|---|---|---|---|---|
| Number of Samples | 2.93e+5 | 2.01e+5 | 4.02e+5 | 8.04e+5 |

Our work replaces these options with a more flexible goal-conditioned control policy. Unlike using an option to reach one goal, the goal-conditioned policy learns to reach any given goal. Because we train only one control policy, all the samples we during training are used for training this policy, while the samples in the options framework are distributed to train multiple policies. Over 5 runs, the average number of samples required to train policies reaching the convergence reward on the Point robot is in Table 6.

## L  Soft and Hard STL Lagrangian Terms

Table 7: Soft and Hard STL Constraints

| | Soft STL | Hard STL |
|---|---|---|
| Seq | **268.20** ± 80.45 | 710.20 ± 193.76 |
| Cover | 1090.60 ± 428.56 | **1035.00** ± 370.58 |
| Branch | 49.00 ± 24.23 | **43.20** ± 9.74 |
| Loop | 119.20 ± 23.07 | **112.60** ± 35.57 |
| Signal | **352.40** ± 289.79 | 588.00 ± 128.79 |

We provided smooth STL Leung et al. (2020; 2022); Jha et al. (2019); Gilpin et al. (2021) with soft min and max in our implementation. In a constrained learning context, STL is used as part of a loss function. To explain the role of soft and hard STL in this context, we conducted an ablation study

showing the efficiency of soft and hard STL loss. We trained our planning policy network with both soft and hard STL loss, and stop training when the policy could generate $95\%$ of paths satisfying the STL specification. Each cell in Table 7 is the result of 5 runs. The mean and stand deviation are before and after $\pm$, resp. The results in Table 7 are mixed. Soft STL loss performs significantly better on task seq and signal, but slightly worse on the cover, branch, and loop tasks on average. Intuitively, choosing soft and hard STL is analogous to choosing $L_2$ or $L_\infty$ loss ($L_2$ loss based on MSE is "smooth" while $L_\infty$ loss based on max is "hard").

