# OpenReview forum: "Constrained Hierarchical Deep Reinforcement Learning with Differentiable Formal Specifications"
_ICLR.cc/2023/Conference — Submitted to ICLR 2023_

### Official Review · Reviewer_Pa7M · 2022-10-24

**Confidence:** 5
**Correctness:** 2
**Technical Novelty And Significance:** 1
**Empirical Novelty And Significance:** 2
**Recommendation:** 3

**Clarity, Quality, Novelty And Reproducibility:**

The paper is well-written, though it is heavy on just restating widely known concepts in formal methods and control. While there is some value in transferring knowledge from one community to another, the use of temporal logic in DRL is widely recognized in the learning community, and it is not useful to just re-summarize basic concepts around temporal logic and its quantitative semantics in the paper. A reference to a book chapter would be sufficient. Similarly, the use of Lagrangian is textbook and doesn't merit the detailed discussion. The space could be better used in describing some novel contribution in the paper.

Please see the weaknesses for further comments on clarity and novelty.



**Strength And Weaknesses:**

Strengths:

- The paper addresses an important problem of using formal/logical specifications for reward shaping in RL

Weaknesses:

- The paper makes a number of claims to novelty that are not accurate when one takes the vast literature on controller synthesis and learning in formal methods, robotics and control literature. The abstraction mentions that "Prior work, however, has failed to consider the possibility of making these specifications differentiable, which would yield a more informative signal of the objective via the specification gradient." This fundamental claim to novelty is inaccurate. Smooth representations of formal specifications (in non-RL) setting have been widely investigated earlier. See for examples: http://robotics.cs.rutgers.edu/wafr2020/wp-content/uploads/sites/7/2020/05/WAFR_2020_FV_55.pdf,  https://link.springer.com/article/10.1007/s10703-019-00332-1 and https://ieeexplore.ieee.org/document/9114883 . When there exists a systematic compilation approach to achieve smoothness, it is important to compare the new approach for differentiability with these existing techniques. The paper needs to provide some justification based on empirical runtime or optimization effectiveness. The reviewer recognizes that this is a new context - DRL, but what is the added challenge in DRL when it comes to having smooth specifications.

- The idea of using a Lagrangian for regularization with constraints is quite common, and it is not clear if the use of temporal logic constraints creates any new challenge.

- The integration of neural network predicates with logic is also widely used. CLIP system from 90s and its recent resurgence in the form of Logic Tensor Networks (LTN): https://www.sciencedirect.com/science/article/abs/pii/S0004370221002009 is an example. Several other mixtures of deep learning and logic use it, including https://proceedings.neurips.cc/paper/2018/hash/dc5d637ed5e62c36ecb73b654b05ba2a-Abstract.html . If the use of this idea in DRL (as against supervised learning applications of LTN) would have involved some special challenge, this claim to novelty would have been justified. At least discussion of these and acknowledgement, the novelty is in adapting a well-known idea to the new setting of DRL would be a more accurate representation.

- It is not clear why the paper views hierarchical decomposition as a novelty. Doesn't the use of temporal logic specification in this paper (and all previous papers doing so and cited in this paper) also implicitly or explicitly end up being hierarchical because temporal logic specifications are compositional? Section 3 does not elaborate on this claim to hierarchy. One is left to assume that the paper is referring to the standard practice of having two steps - planning and control.

- The section on differentiable specification is rather vague, and it is unclear what is the algorithm here (making it difficult to compare conceptually with the literature mentioned above).
 -- The paper points to the neural signed distance function as an example of a differentiable predicate, and then just states that "our approach .. is not limited to neural SDFs". What other neural predicates can be learned?
 -- The mention of IRL and claim that they can learn differentiable predicate in this context is confusing. The entire claim in the paper is of the use of temporal logic specifications and hierarchical planning/policy generation, why would IRL to learn reward functions be relevant here? The discussion goes on to say the paper is "not on investigating the diversity and application of neural predicates." which may well be true but one needs more than one example of neural predicates. Also, how these would form a differentiable temporal logic is left vague.
 -- The use of max/min for globally/future in Equation 2/3 are just the standard quantitative semantics of temporal logic that is widely known. If the entire claim to differentiability rests on using this well-known quantitative semantics, then it is alarming because it is widely known that this semantics (due to max/min) is in practice not very smooth (and is the motivation of the literature mentioned in the review above).

- Can authors discuss the similarity and differences in their use of neural-ODE and the TrajODE https://www.ijcai.org/proceedings/2021/0207.pdf  (both conceptually and empirically)?

- The experimental evaluation uses very simple environments, and all of them are essentially path planning examples. Doing an ablation study, where existing smooth representation of temporal logic are considered, would help better explain the value of the proposed approach.

**Summary Of The Paper:**

The paper builds on recent work in formal methods and ML community on using temporal logic specifications to shape rewards for deep reinforcement learning. The main claim to novelty is the use of differentiable representation of temporal logic specifications. A secondary claim is hierarchical setting where path planner works with low-level control policy.

**Summary Of The Review:**

The paper appears to identify some of the most important needs for making DRL work in practice - hierarchical policies, and the use of temporal logic specification. It seems to be ignoring literature over the last few years on a related topic and makes claims to novelty that are not justified. Further, the presentation of the approach is vague and most of the paper is just re-summarizing known concepts from formal methods and control. Perhaps, presenting this work as taking ideas from different sub-areas and adapting them to DRL would have been much better. But in that case, the paper needs to justify the extra challenges in adapting to DRL and use more involved examples (beyond simple planning/control examples) to show the benefit.

---

> ### Author Response · Authors · 2022-11-19
> **Clarifications about Reviews and Our Contribution**
>
> We would like to express our deepest gratitude for your constructive feedback again! We have added a comparative discussion to these papers in our revision. However, we believe there exist several misunderstandings regarding our contributions that we clarify below.
>
>  *  **Our main contribution is a more tractable DRL learning framework with LTL than existing approaches.** Although we build our work on several existing techniques, the novelty of their integration leads to a new framework with superior performance and capabilities compared to existing work, which is demonstrated with extensive empirical study.
>  * **We agree with reviewers KU3V and Ea5U who said the benchmarks are "rather challenging" (reviewers KU3V) and "superior to other methods in this area" (reviewers Ea5U).** The ability to deal with this complexity is a result of co-optimizing the high-level plan policy with the low-level control policy on the unknown, high-dimensional (up to 58 dimensions), stochastic, non-linear dynamics. To the best of our knowledge, no existing work in this space (LTL-guided DRL learning) has scaled to such high dimensional benchmarks. Indeed, all of the papers cited by the reviewer consider a significantly simple setup than we do (e.g., low-dimensional and known dynamics). We provide additional discussion below.
>  * **The key factor that makes our framework outperform existing reward-shaping works is improved sample complexity.**  The key here is that we do not force constraint satisfaction with reward shaping and a sample-based policy gradient.  Instead, constraint satisfaction is achieved by integrating STL as a differentiable Lagrangian term, which is solved with a dual ascent algorithm. The benefits of doing so are elaborated below.
>  * **Integrating LTL to constrain the neural network policy updates poses new challenges.** Our goal is to constrain the training of a neural network policy with formal specifications. Logical-solver-based approaches can indeed constrain the planned path, but they cannot constrain the training of neural network policy directly in the parameter space. The solver may generate a path and constrain the policy output with a runtime shield (as mentioned by reviewer KU3V) in the path space. However, because constraints on the path space cannot affect the parameter space directly, it is hard to integrate such techniques with gradient-based learning systems. That is why we use the Lagrangian term, where the gradient computed with the specification can directly constrain the parameter updates of neural network policy.
>  * **We do not claim that backpropagating through STL is our novelty.** To address the concern of lacking discussion of papers pointed out by the reviewer, our revision expands the discussion in related work to include these papers; a detailed comparison is also provided below.  We also moved the backpropagating through STL to the background section.
>  * **The context when we say previous work fails to make these specifications differentiable is after the discussion on DRL using LTL reward shaping.** The scope is in the DRL with LTL instructions. We have made this clear in the revision.
>  * **We do not view the use of hierarchical policies as a novel contribution.** We discuss both existing hierarchical DRL and hierarchical LTL reward-shaping work in the third paragraph of the introduction. Although the hierarchical setting is not new, the formulation of a goal-conditioned policy indeed differentiates our work from existing approaches that use LTL in a DRL setting.
>
> We realize that the different communities might have different focal points of interest; we provide additional elaboration on the reviewer's comments below and frame our contributions in light of these concerns below.

---

> > ### Author Response · Authors · 2022-11-19
> > **Existing STL Planning Work**
> >
> > ## Existing STL Planning Work
> > The system dynamics in our work are unknown (model-free), high-dimensional (up to 58 dimensions in our paper), stochastic, and non-linear. These challengings make traditional STL planning [1,2] unable to be applied to our benchmarks with the dynamics and controller constraints.
> >
> > However, considering the dynamics constraints and controller capabilities is crucial when learning a planning policy. Our framework addresses this challenge by co-optimizing the planning and control policy in the same loop.
> >
> > Furthermore, because we train a neural network as a planning policy, we can generate planned paths faster than traditional STL planning approaches like [1,2].
> >
> > ### Dynamics
> > Below are detailed differences between our settings and [1, 2]:
> >
> > * They assume the system dynamics are known and deterministic, while our system dynamics are unknown and stochastic.
> > * The state space of the system dynamics is up to 2 dimensions in [1, 2], while our system dynamics have state spaces of up to 58 dimensions.
> > * One example in [1] considered a 2D unicycle non-linear system dynamics, all the other cases in [1, 2] are linear systems. In contrast, all the dynamics systems we considered are high dimensional (state dimensions range from 14 to 58), non-linear and underactuated.
> >
> > Even discarding all the other difficulties, the unknown system dynamics setting makes it impossible to integrate the dynamics constraints as part of an STL specification and co-optimize them with gradient/MILP/SQP like [1, 2].
> >
> > In the DRL with LTL context, more related papers (e.g., [3, 4, 5]) have been discussed both conceptually and empirically.
> >
> > ### Co-Optimizing Planning and Control Policy
> > It is possible to ignore the dynamics and directly do path planning with STL, and subsequently combine the planned path with all the other components in our framework. However, as we demonstrate in Sec. 4.4.1, co-learning the planning and control policies are critical for performance, because the control and planning policies adapt to each other as part of the co-learning process.
> >
> > The performance decay while using a gradient-based planner is discussed in 4.4.1, where the smooth gradient is used for generating the path.
> >
> > ### Planning Time
> > One benefit we mentioned in Sec. 4.4.1 is planning time. Compared with [1,2], our approach requires less planning time when deployed online.
> >
> > The planning policy is a neural network that generates planning paths from randomly sampled initial states. A forward propagation in milliseconds will produce a path taking MIP [1,9] or gradient technique 2 for several seconds to solve. We quantify the planning time below.
> >
> > | Tasks  | Planning Policy | MIP      | Gradient  |
> > | ------ | --------------- | -------- | --------- |
> > | Seq    | **0.001722**    | 0.096137 | 0.842844  |
> > | Cover  | **0.00177**     | 0.218563 | 3.484746  |
> > | Branch | **0.001437**    | 0.078942 | 7.204325  |
> > | Loop   | **0.004844**    | 7.423811 | 10.480951 |
> > | Signal | **0.002383**    | 7.853121 | 2.670637  |
> >
> > The time is the average time over 100 runs. Our planning policy is at least 55X faster than MIP planners and 489X faster than gradient-based planners.
> >
> > ## Smooth STL
> > We provided smooth (a.k.a soft) min and max in our implementation. The gradient planner we evaluated in 4.4.1 is based on smooth STL.
> >
> > In a constrained learning context, STL is used as part of a loss function. To explain the role of soft and hard STL in this context, we conducted an ablation study showing the efficiency of soft and hard STL loss. We train our planning policy network with both soft and hard STL loss, and stop training when the policy can generate 95% of paths satisfying the STL specification. Each cell in the following table is the result of 5 runs. The mean and std are before and after ±, resp.
> >
> > | Tasks  | Soft STL            | Hard STL             |
> > | ------ | ------------------- | -------------------- |
> > | seq    | **268.20** ± 80.45  | 710.20 ± 193.76      |
> > | cover  | 1090.60 ± 428.56    | **1035.00** ± 370.58 |
> > | branch | 49.00 ± 24.23       | **43.20** ± 9.74     |
> > | loop   | 119.20 ± 23.07      | **112.60** ± 35.57   |
> > | signal | **352.40** ± 289.79 | 588.00 ± 128.79      |
> >
> > The results are mixed. Soft STL loss performs significantly better on task seq and signal, but slightly worse on the cover, branch, and loop tasks on average. Intuitively, choosing soft and hard STL is analogous to choosing $L_{2}$ or $L_{\infty}$ loss ($L_{2}$ loss based on MSE is "smooth" while $L_{\infty}$ loss based on max is hard).

---

> > > ### Author Response · Authors · 2022-11-19
> > > **A More Tractable Framework**
> > >
> > > We hope to highlight that our most significant contribution is a tractable hierarchical reinforcement learning framework scaled to challenging benchmarks that have not been solved by existing work ([3, 4, 5]). Two reasons make our framework more tractable.
> > >
> > > 1. We discuss sample complexity issues of LTL reward shaping in the first paragraph of the introduction. A more theoretical discussion is provided in [6]. The main argument here is that the accurate policy gradient estimated from samples requires a large number of samples, while the gradient computed from the STL specification is accurate and requires many fewer samples in the aggregate.
> > > 2. While we only learn one goal-conditioned policy, existing hierarchical LTL techniques learn multiple option policies, as noted in the third paragraph of the introduction. This scales poorly as specification complexity increases.
> > >
> > > These two claims are well-supported in our comparisons to these previous reward-shaping frameworks in Fig.3 and Fig.12. We have conducted two more ablation experiments (given below) to justify them further.
> > >
> > > ### Planning Layer Sample Complexity
> > > To better demonstrate the sample complexity issue in reward shaping, we run both the policy gradient algorithm and policy gradient constrained by STL specifications on a path planning problem.
> > >
> > > | Tasks  | Policy Gradient          | Constrained Policy Gradient      |
> > > | ------ | ------------------------ | -------------------------------- |
> > > | Seq    | 9800.00 ± 0.00 (1 / 5)   | **640.00** ± 101.98 (**5** / 5)  |
> > > | Cover  | N/A (0 / 5)              | **780.00** ± 584.47 (**5** / 5)  |
> > > | Branch | 9700.00 ± 200.00 (2 / 5) | **720.00** ± 526.88 (**5** / 5)  |
> > > | Loop   | N/A (0 / 5)              | **1240.00** ± 149.67 (**5** / 5) |
> > > | Signal | N/A (0 / 5)              | **1500.00** ± 419.52 (**5** / 5) |
> > >
> > > We run both the policy gradient ([3, 4, 5]) and constrained policy gradient  (ours) algorithm 5 times within 10000 gradient updates. The gradient updates will stop if over 95% of planned paths satisfy the STL specification. In each column, the number before and after ± is the mean and standard deviation of gradient updates, resp. N/A means in 5 runs; the algorithm failed to train a planning policy with over 95% satisfaction rate. The 2 numbers in the parentheses are the runs successfully trained a policy and the total runs, resp.
> > >
> > > In all the 5 tasks evaluated, constrained policy gradient (with the Lagrangian term) results in 10x fewer gradient steps and can always successfully train a policy.
> > >
> > > ### Goal-Conditioned Control Policy
> > > Previous work learns one control policy (option) for each transition in the automaton built with an LTL specification. For example, a sequence task with four goals will require four policies.
> > >
> > > Our work replaces these options with a more flexible goal-conditioned control policy. Unlike using an option to reach one goal, the goal-conditioned policy learns to reach any given goal. Because we train only one control policy, all the samples we during training are used for training this policy, while the samples in the options framework are distributed to train multiple policies. Over 5 runs, the average number of samples required to train policies reaching the convergence reward on the Point robot is below.
> > >
> > > |            | Goal-Conditioned Policy | 1 Option | 2 Options | 4 Options |
> > > | ---------- | :---------------------: | :------: | :-------: | :-------: |
> > > | #. Samples |         2.93e5          |  2.01e5  |  4.02e5   |  8.04e5   |
> > >
> > > Although training a goal-conditioned policy requires more samples than training 1 option, because the options are designed for different tasks with different rewards, the aggregate number of samples increases linearly as the number of options increases. As tasks become more complicated (e.g., if they need to follow a long sequence), an option-based approach becomes correspondingly harder (or impossible) to train.

---

> > > > ### Author Response · Authors · 2022-11-19
> > > > **Miscellaneous and Reference**
> > > >
> > > > ## Misc.
> > > > * We introduce neural predicates because we want to show that our framework can be extended with any quantitative functions on states. We mention IRL because it learns a reward function on a state. We have elaborated on this motivation in the revised version of the paper.
> > > > * As an IRL example, an expert drone operator dataset can be used to learn reward functions measuring a policy's similarity to expert behaviors. Integrating this similarity reward function into an STL specification, and constraining policy with this specification will force the policy to behave similarly to the expert. There also exist multiple expert datasets, since the use of offline RL that relies on such datasets has become increasingly popular.
> > > > * The neural predicate work [7,8] pointed out by the reviewer is not applied in conjunction with STL. We have discussed these papers in the revision. Representing programs with neural networks is indeed a very wide field. [10] mentioned by the reviewer is in the STL context, but it does not focus on learning neural network predicate.
> > > > * TrajODE applies the ODE solver in the latent embedding space, while neural ODE applies the ODE solver directly in the state space. In terms of their functionality in our framework, they are similar, because both of them can backpropagate efficiently on a trajectory, and have a memory about the past. The authors of TrajODE do not open-source their code. However, if the reviewer is interested in the comparison, we can implement their code as an additional experiment.
> > > > * To address the concern that backpropagation through STL is not our contribution, we have moved this description to the background section.
> > > >
> > > > *The above data has already been added to the appendix or paper. All the changes in the revision are highlighted. The code for all experiments generated for the rebuttal is also available in the anonymous repository in the replication statement.*
> > > >
> > > >
> > > > **Reference**
> > > > [1] Pant, Yash Vardhan, Houssam Abbas, and Rahul Mangharam. "Smooth operator: Control using the smooth robustness of temporal logic." 2017 IEEE Conference on Control Technology and Applications (CCTA). IEEE, 2017.
> > > >
> > > > [2] Leung, Karen, Nikos Aréchiga, and Marco Pavone. "Back-propagation through signal temporal logic specifications: Infusing logical structure into gradient-based methods." International Workshop on the Algorithmic Foundations of Robotics. Springer, Cham, 2020.
> > > >
> > > > [3] Icarte, Rodrigo Toro, et al. "Using reward machines for high-level task specification and decomposition in reinforcement learning." International Conference on Machine Learning. PMLR, 2018.
> > > >
> > > > [4] Hasanbeig, Mohammadhosein, et al. "Reinforcement learning for temporal logic control synthesis with probabilistic satisfaction guarantees." 2019 IEEE 58th Conference on Decision and Control (CDC). IEEE, 2019.
> > > >
> > > > [5] Jothimurugan, Kishor, et al. "Compositional reinforcement learning from logical specifications." Advances in Neural Information Processing Systems 34 (2021): 10026-10039.
> > > >
> > > > [6] Yang, Cambridge, Michael Littman, and Michael Carbin. "On the (In)Tractability of Reinforcement Learning for LTL Objectives" IJCAI 2021.
> > > >
> > > > [7] Samy Badreddine, Artur d'Avila Garcez, Luciano Serafini, Michael Spranger, "Logic Tensor Networks"
> > > >
> > > > [8] Manhaeve, Robin, et al. "Deepproblog: Neural probabilistic logic programming." Advances in Neural Information Processing Systems 31 (2018).
> > > >
> > > > [9] Dawei Sun, Jingkai Chen, Sayan Mitra, Chuchu Fan, "Multi-agent Motion Planning from Signal Temporal Logic Specifications" IEEE Robotics and Automation Letters (RA-L) (2022)
> > > >
> > > > [10] Susmit Jha, Ashish Tiwari, Sanjit A. Seshia, Tuhin Sahai & Natarajan Shankar, "TeLEx: learning signal temporal logic from positive examples using tightness metric", Formal Methods in System Design (2019)

---

> > > > > ### Comment · Reviewer_Pa7M · 2022-11-30
> > > > > **"novelty of integration" needs to be better explained and prior work clearly demarcated**
> > > > >
> > > > > Adding comment here if the one above is not visible. A major rewrite of the paper that removes some of the prior work discussion and points to existing literature (identified in this discussion thread) or clearly demarcates what is new in this paper compared to state-of-the-art would make the paper more transparently communicate its contributions.
> > > > >
> > > > > --
> > > > >
> > > > > The reviewer appreciates clarification that some of the ideas are not novel, and the detailed discussion of related work in the response would be a very useful addition to the paper to make sure all readers get enough context about past work. This will ensure all readers have a clear idea of how the current draft builds on the existing state-of-the art.
> > > > >
> > > > > > Although we build our work on several existing techniques, the novelty of their integration leads to a new framework with superior performance and capabilities compared to existing work, which is demonstrated with extensive empirical study.
> > > > >
> > > > > The original presentation with the missing references did not give this perception that the novelty is limited to integration of existing techniques into a new "framework", and it is likely that the readers less familiar with the enormous amount of work on STL, RL and controller synthesis would overestimate the contribution of this paper. What is still missing is identification of what is novel and non-trivial about this integration? How do techniques that use a Lagrangian term corresponding to an arbitrary logical formula need to be extended in a non-trivial way when using STL and its known smooth quantitative semantics?
> > > > >
> > > > > > Instead, constraint satisfaction is achieved by integrating STL as a differentiable Lagrangian term, which is solved with a dual ascent algorithm.
> > > > >
> > > > > Using Langrangian to integrate logical constraints and the use of dual descent is an old idea. For example, see https://arxiv.org/pdf/1805.07075.pdf . For this formulation, there is no additional challenge posed by what fragment of logic is used to express the property - whether it is STL, LTL, CTL or any other fragment.
> > > > >
> > > > > A premier venue expects a paper to have significant novelty - currently, the draft is incrementally extending ideas where the extensions and integrations are not particularly challenging. Conflation of background explaining prior work and what is new, needs to be avoided.

---

> ### Author Response · Authors · 2022-11-29
> **Any remaining questions?**
>
> Dear reviewer,
>
> Please let us know if our previous response addresses your questions. We are happy to answer any remaining questions.

---

> ### Comment · Reviewer_Pa7M · 2022-11-30
> **Not sufficient novelty**
>
> The reviewer appreciates clarification that some of the ideas are not novel, and the detailed discussion of related work in the response would be a very useful addition to the paper to make sure all readers get enough context about past work. This will ensure all readers have a clear idea of how the current draft builds on the existing state-of-the art.
>
> > Although we build our work on several existing techniques, the novelty of their integration leads to a new framework with superior performance and capabilities compared to existing work, which is demonstrated with extensive empirical study.
>
> The original presentation with the missing references did not give this perception that the novelty is limited to integration of existing techniques into a new "framework", and it is likely that the readers less familiar with the enormous amount of work on STL, RL and controller synthesis would overestimate the contribution of this paper. What is still missing is identification of what is novel and non-trivial about this integration? How do techniques that use a Lagrangian term corresponding to an arbitrary logical formula need to be extended in a non-trivial way when using STL and its known smooth quantitative semantics?
>
> > Instead, constraint satisfaction is achieved by integrating STL as a differentiable Lagrangian term, which is solved with a dual ascent algorithm.
>
> Using Langrangian to integrate logical constraints and the use of dual descent is an old idea. For example, see https://arxiv.org/pdf/1805.07075.pdf . For this formulation, there is no additional challenge posed by what fragment of logic is used to express the property - whether it is STL, LTL, CTL or any other fragment.
>
> A premier venue expects a paper to have significant novelty - currently, the draft is incrementally extending ideas where the extensions and integrations are not particularly challenging. Conflation of background explaining prior work and what is new, needs to be avoided.

---

> > ### Author Response · Authors · 2022-12-01
> > **Response to the remaining concerns (1/2)**
> >
> > We thank the reviewer again for responding to our rebuttal and presenting the remaining concerns.  We respectfully disagree with the reviewer's conclusion that the integration of techniques developed for different settings limits the novelty of our approach. We elaborate on our reasons below.
> >
> > > For this formulation, there is no additional challenge posed by what fragment of logic is used to express the property - whether it is STL, LTL, CTL or any other fragment.
> >
> > **No undifferentiable formula (e.g., LTL, CTL) can be integrated as a Lagrangian multiplier in a loss function.** Training a neural network policy requires backpropagating the gradient from the loss function and updating parameters with the gradient. If the specification is not differentiable, it cannot generate a gradient and will not affect the training of neural network policies.
> >
> > There have been many Lagrangian-based techniques that manifest as a *single* descent/ascent algorithm by setting a *fixed* large $\lambda$ (e.g., https://arxiv.org/pdf/1805.07075.pdf highlights that their $\lambda$ is *set*, and "can set a *very high value*"). A fixed large $\lambda$ typically leads to overly weighting constraints but ignores performance objectives in the loss function. The performance part of our *planning* policy optimizing on *control* reward is important because it adapts the planning policy to the control policy. The importance of such adaptation is illustrated with detailed ablation in 4.4.1. It is observations such as this that form the basis for our claims of novelty.
> >
> > The dual ascent algorithm will dynamically change $\lambda$ based on the "degree of satisfaction" provided by STL. When the degree of satisfaction is high, $\lambda$ will be decreased and the optimization will focus on the performance part. Otherwise, it will be increased and will focus on satisfying constraints defined by the STL formula. This is another reason why STL providing the degree of satisfaction is necessary, because CTL, LTL, etc. do not provide such measurements.
> >
> > The **Lagrangian Methods in Constrained DRL** section in Background explained that the dual ascent algorithm and the constraint function must be differentiable.

---

> > > ### Author Response · Authors · 2022-12-01
> > > **Response to the remaining concerns (2/2)**
> > >
> > > >  What is still missing is identification of what is novel and non-trivial about this integration? How do techniques that use a Lagrangian term corresponding to an arbitrary logical formula need to be extended in a non-trivial way when using STL and its known smooth quantitative semantics?
> > >
> > > **Combining differentiable STL with DRL has never been explored in previous work**. Previous reward-shaping techniques suffer from sample complexity limitations and the multi-option issues have been highlighted at the beginning of the introduction (in the original version of our submission); these points are further elaborated with more ablation studies in our revised version. We reiterate that our claim of novelty is based on our ability to overcome sample complexity issues in a DRL setting (as evidenced by our extensive experimental evaluation).
> > >
> > > The paper mentioned by the reviewer discusses STL differentiability, and which is provided as evidence on the lack of novelty of our approach, operates on known, low-dimensional, deterministic, and linear systems. **Notably, the cited paper does not (and fundamentally cannot) leverage STL's differentiability to more complicated systems, such as the ones we consider which involve high-dimensional, unknown, stochastic, non-linear and underactuated features**.
> > >
> > > An important distinction between the various papers mentioned by the reviewer and our technique is that the reviewer's points of comparison **do not** relate to DRL and none of the cited papers demonstrate **scalability** to high-dimensional, unknown, stochastic, non-linear and underactuated systems. The challenge of combining DRL with symbolic instructions is extensively discussed in the introduction, experiments, and related work.  As explained above, the challenge in this setting comes from the sample complexity of the policy gradient. We have provided an extensive comparison with existing reward-shaping work to demonstrate the above points in Sec. 4.2 and Appendix H. The sample complexity issue is also further explained with an additional ablation study in Sec. 4.4.3.
> > >
> > > **Beyond the integration of STL, techniques such as integrating neural ODE policies, replacing options with a goal-conditioned policy, and the use of a neural SDF in this setting are non-trival, and have also not been explored in previous DRL work with symbolic specifications**. These non-trivial integrations of these features are also critical to achieve scalability and provide a tractable solution for dealing with sample complexity.
> > >
> > > Beyond the contributions and novelty of our framework, we do not believe the extensive empirical results we provide, which demonstrate scalability that has not been previously shown, should be summarily discounted.  (As an aside, we point out that all of our code and models are public, providing a set of benchmarks for future work.)
> > >
> > > > Discussion of related work in the response
> > >
> > > We summarize the differences between the various papers cited by the reviewer and our work. In general, these cited papers focus on problems and settings that are vastly different from ours. These differences can be summarized into the following categories, and we link them with the papers following. The related discussions have also been provided in our revision.
> > >
> > > (1) The work does not relate to DRL and thus does not address fundamental challenges to sample complexity and policy training.
> > > (2) The work considers open-loop or behavior cloning problems.
> > > (3) The work fundamentally cannot scale to the application we consider (unknown, high-dimensional, non-linear, stochastic).
> > >
> > > - [Back-propagation through Signal Temporal Logic Specifications: Infusing Logical Structure into Gradient-Based Methods](http://robotics.cs.rutgers.edu/wafr2020/wp-content/uploads/sites/7/2020/05/WAFR_2020_FV_55.pdf) (1,3)
> > > - [A Smooth Robustness Measure of Signal Temporal Logic for Symbolic Control](https://ieeexplore.ieee.org/document/9114883) (1,3)
> > > - [Logic Tensor Networks](https://www.sciencedirect.com/science/article/abs/pii/S0004370221002009) (1,2)
> > > - [DeepProbLog: Neural Probabilistic Logic Programming](https://proceedings.neurips.cc/paper/2018/hash/dc5d637ed5e62c36ecb73b654b05ba2a-Abstract.html) (1,2)
> > > - [Trusted Neural Networks for Safety-Constrained Autonomous Control](https://arxiv.org/pdf/1805.07075.pdf) (1,2)
> > > - [TeLEx: learning signal temporal logic from positive examples using tightness metric](https://link.springer.com/article/10.1007/s10703-019-00332-1) (1,2)

---

### Official Review · Reviewer_Ea5U · 2022-10-25

**Confidence:** 3
**Correctness:** 4
**Technical Novelty And Significance:** 2
**Empirical Novelty And Significance:** 2
**Recommendation:** 5

**Clarity, Quality, Novelty And Reproducibility:**

The proposed approach it clearly laid out and the ablation seems sufficiently thorough.

The technique builds off existing components in the literature, but does so in a well motivated way.

**Strength And Weaknesses:**

# Strength

The developed algorithm provides a well-motivated construction of a constrained policy gradient algorithm by applying lagrangian methods to a STL formula's quantitative semantics. The result seems technically sound and (in my opinion) superior to other methods I have seen in this area.

# Weakness

That said, I am suspect of the novelty of using a differentiable STL formula to guide controller synthesis. While I am not deeply involved in this area, I know of at least two works [1],[2] with a quick google scholar search revealing many related articles. I view the comparison with these existing methods.

If the authors could clarify why these are either not appropriate baselines or provide a comparison, I would be willing to increase my score.

Finally, as with all such methods, it seems that the "units" of the STL formula are important for determining the slope of the constraint function. An analysis of changing the STL formula to have the same qualitative semantics, but different quantitative semantics would have been appreciated, i.e., by changing the predicates. The result would have a different gradient landscape.

[1] Pant, Yash Vardhan, Houssam Abbas, and Rahul Mangharam. "Smooth operator: Control using the smooth robustness of temporal logic." 2017 IEEE Conference on Control Technology and Applications (CCTA). IEEE, 2017.
[2] Leung, Karen, Nikos Aréchiga, and Marco Pavone. "Back-propagation through signal temporal logic specifications: Infusing logical structure into gradient-based methods." International Workshop on the Algorithmic Foundations of Robotics. Springer, Cham, 2020.

**Summary Of The Paper:**

This paper treats the problem of constrained deep reinforcement learning, where constraints are given in the form of signal temporal logic (STL). The paper proposes a (to my knowledge) novel technique for using Lagrangian methods with STL. They evaluate on several domains and illustrate the efficacy of their approach.

**Summary Of The Review:**

My primary concern with this paper is the lack of comparison with other differentiable STL methods that have popped up in the literature. While I believe this approach is well motivated despite this, an evaluation or discussion would help set this paper in context.

---

> ### Author Response · Authors · 2022-11-19
> **Response to Reviewer Ea5U (1/2)**
>
> We would like to express our deepest gratitude for your constructive feedback again! We thank the reviewer for pointing out related work in the context of LTL-guided controller synthesis.
>
> + We have revised our manuscript in response to discuss the related LTL-guided controller synthesis work.
> + The difference, connection, and comparison to these papers are detailed below.
> + We hope reviewer can clarify the question regarding the units of the STL formula.
>
> ## LTL-guided Controller Synthesis Baselines
> We believe references [1, 2] are not appropriate baselines because they make a completely different assumption on the availability and characteristics of the underlying system dynamics. The scope of our paper is controller (policy) synthesis with *model-free* deep reinforcement learning, where the dynamics are unknown, high-dimensional (up to 58 dimensions in our paper), stochastic, and non-linear. In this context, more related papers (e.g., [3, 4, 5]) have been discussed both conceptually and empirically in the paper.
>
> To elaborate, some of the salient differences between our setting and [1,2] are:
> * They assume system dynamics are known and deterministic, while our system dynamics are unknown and stochastic.
> * The state space of the system dynamics considered in [1,2] is up to 2 dimensions, while our system dynamics have state spaces of up to 58 dimensions.
> * One example in [1] considered a 2D unicycle non-linear system dynamics; all the other cases in [1,2] are linear systems. In contrast,  the dynamics of the systems we considered are non-linear and underactuated.
>
> The difficulty and opacity of our system dynamics make it hard or impossible to integrate the underlying dynamics as part of an STL specification and to have them solved with gradient/MILP/SQP like [1, 2].
>
> ### Co-Optimizing Planning and Control Policy
> It is possible to ignore the dynamics and directly do path planning with STL, and subsequently combine the planned path with all the other components in our framework. However, as we demonstrate in Sec. 4.4.1, co-learning the planning and control policies are critical for performance, because the control and planning policies adapt to each other as part of the co-learning process.
>
> The performance decay while using a gradient-based planner is discussed in 4.4.1, where the smooth gradient is used for generating the path.
>
> ### Planning Time
> One benefit we mentioned in Sec. 4.4.1 is planning time. Compared with [1,2], our approach requires less planning time when deployed online.
>
> The planning policy is a neural network that generates planning paths from randomly sampled initial states. A forward propagation in milliseconds will produce a path taking MIP [1] or gradient techniques for several seconds to solve. We quantify the planning time below.
>
> | Tasks  | Planning Policy | MIP      | Gradient  |
> | ------ | --------------- | -------- | --------- |
> | Seq    | **0.001722**    | 0.096137 | 0.842844  |
> | Cover  | **0.00177**     | 0.218563 | 3.484746  |
> | Branch | **0.001437**    | 0.078942 | 7.204325  |
> | Loop   | **0.004844**    | 7.423811 | 10.480951 |
> | Signal | **0.002383**    | 7.853121 | 2.670637  |
>
> The time is the average time over 100 runs. Our planning policy is at least 55X faster than MIP planners and 489X faster than gradient-based planners.
>
> ### Smooth STL
> We provided smooth (a.k.a soft) min and max [1,2] in our implementation. The gradient planner we evaluated in 4.4.1 is based on smooth STL.
>
> In a constrained learning context, STL is used as part of a loss function. To explain the role of soft and hard STL in this context, we conducted an ablation study showing the efficiency of soft and hard STL loss. We trained our planning policy network with both soft and hard STL loss, and stop training when the policy can generate 95% of paths satisfying the STL specification. Each cell in the following table is the result of 5 runs. The mean and std are before and after ±, resp.
>
> | Tasks  | Soft STL            | Hard STL             |
> | ------ | ------------------- | -------------------- |
> | seq    | **268.20** ± 80.45  | 710.20 ± 193.76      |
> | cover  | 1090.60 ± 428.56    | **1035.00** ± 370.58 |
> | branch | 49.00 ± 24.23       | **43.20** ± 9.74     |
> | loop   | 119.20 ± 23.07      | **112.60** ± 35.57   |
> | signal | **352.40** ± 289.79 | 588.00 ± 128.79      |
>
> The results are mixed. Soft STL loss performs significantly better on task seq and signal, but slightly worse on the cover, branch, and loop tasks on average. Intuitively, choosing soft and hard STL is analogous to choosing $L_{2}$ or $L_{\infty}$ loss ($L_{2}$ loss based on MSE is "smooth" while $L_{\infty}$ loss based on max is hard).

---

> > ### Author Response · Authors · 2022-11-19
> > **Response to Reviewer Ea5U (2/2)**
> >
> > ##  Units of the STL Formula
> > Can the reviewer elaborate a bit on the concern related to "have the same qualitative semantics, but different quantitative semantics"?
> >
> > If the reviewer is asking us to keep the TLTL/STL formula, and change the predicate, this is infeasible. As defined in the grammar of TLTL, the predicate $\mathcal{P}$ is a part of the TLTL formula. Changing the predicate means changing the TLTL formula.
> >
> > If the reviewer is talking about **the output unit** of predicates, we compressed the output of the predicates to lie between (-1, 1) with a `tanh` in practice.
> >
> > Can the reviewer clarify what the expectation is here? We are willing to add an analysis if the reviewer can clarify their concerns.
> >
> > **Reference**
> > [1] Pant, Yash Vardhan, Houssam Abbas, and Rahul Mangharam. "Smooth operator: Control using the smooth robustness of temporal logic." 2017 IEEE Conference on Control Technology and Applications (CCTA). IEEE, 2017.
> >
> > [2] Leung, Karen, Nikos Aréchiga, and Marco Pavone. "Back-propagation through signal temporal logic specifications: Infusing logical structure into gradient-based methods." International Workshop on the Algorithmic Foundations of Robotics. Springer, Cham, 2020.
> >
> > [3] Icarte, Rodrigo Toro, et al. "Using reward machines for high-level task specification and decomposition in reinforcement learning." International Conference on Machine Learning. PMLR, 2018.
> >
> > [4] Hasanbeig, Mohammadhosein, et al. "Reinforcement learning for temporal logic control synthesis with probabilistic satisfaction guarantees." 2019 IEEE 58th Conference on Decision and Control (CDC). IEEE, 2019.
> >
> > [5] Jothimurugan, Kishor, et al. "Compositional reinforcement learning from logical specifications." Advances in Neural Information Processing Systems 34 (2021): 10026-10039.

---

> > > ### Comment · Reviewer_Ea5U · 2022-12-13
> > > **Units of the formula**
> > >
> > > Yes. I was referring to the unit of the output predicates. It's a minor point and not something I view as unique to this work, but the derivative of the STL formula ignores the Lie derivatives that determine how the system can actually evolve. A value may seem close to the boundary, but actually require substantial effort to move it due to the system dynamic, e.g., a very large mass causing large inertia...
> > >
> > > The tanh trick doesn't really remove this due to the chain rule...

---

> > > > ### Author Response · Authors · 2022-12-13
> > > > **Appropriate Baselines, Comparison, and Minor Issue on Units of Formulas**
> > > >
> > > > > If the authors could clarify why these are either *not appropriate baselines* or provide a *comparison*, I would be willing to increase my score.
> > > >
> > > > Have our other responses addressed the concerns on the control synthesis baselines?
> > > >
> > > > We clarified that [1][2] are not appropriate baselines because they focus on known, low-dimensional, deterministic, and linear dynamics, while we study unknown, high-dimensional, stochastic, and non-linear dynamics. The difficulty and opacity of our dynamics make it impossible to solve them with gradient/MILP/SQP like [1][2].
> > > >
> > > > The additional experiments further provided a comparison in planning time and showed that the smoothness problem [1] in gradient planning is not an issue for the training neural network planning policies.
> > > >
> > > > > Yes. I was referring to the unit of the output predicates. It’s a minor point and not something I view as unique to this work, ...
> > > >
> > > > We hope the below responses will address this minor issue.
> > > >
> > > > The unit scale and its normalization issue widely exist in deep learning applications (e.g., one input feature dimension can be very large and dominate the final output). Several tricks can handle this minor issue. An example is the batch normalization layer, which estimates the mean and standard deviation during run time, and uses them to normalize data. Similarly, adding tanh will affect the chaining backpropagation process, because it normalizes the output scales of predicates with different output ranges (note that the tanh is not added to the output of final formula, it is added to each predicate).
> > > >
> > > > > A value may seem close to the boundary, but actually require substantial effort to move it due to the system dynamic, e.g., a very large mass causing large inertia...
> > > >
> > > > Our constrained policy gradient updates on planner policy will take into account the different control rewards due to varying dynamics (eq.(4)). The STL/reward normalization can indeed be an issue, like in most deep learning applications. However, various design choices and tricks (e.g., batch normalization mentioned above) can handle the unit issue.

---

> ### Author Response · Authors · 2022-11-29
> **Any remaining questions?**
>
> Dear reviewer,
>
> Please let us know if our previous response addresses your questions. We are happy to answer any remaining questions.

---

### Official Review · Reviewer_KU3V · 2022-10-27

**Confidence:** 4
**Correctness:** 4
**Technical Novelty And Significance:** 3
**Empirical Novelty And Significance:** 3
**Recommendation:** 6

**Clarity, Quality, Novelty And Reproducibility:**

As outlined in the strengths/weaknesses, I do believe this is a good paper which is worth to be accepted. The quality is high, the presentation and ideas are clear, and the reproducibility is strong via a strong numerical evaluation and the availability of code. To the best of my knowledge, this is a novel approach, but one weakness I see is that the authors should provide a better comparison/discussion with respect to probabilistic constraints. I will list a few papers below, some already included in the submitted version. In the essence, there has been a lot of work on RL or safe RL against/according to temporal constraints, based on MDP semantics, where one defines the satisfaction in a probabilistic way. Eg: At state s, the probability of satisfying an LTL spec is p. Isn’t this also a differentiable notion? If yes, that should be made very clear. The only other comment I have is that I found the introduction to the LTL variant rather longish and mostly standard.

Mohammadhosein Hasanbeig, Yiannis Kantaros, Alessandro Abate, Daniel Kroening, George J Pap- pas, and Insup Lee. Reinforcement learning for temporal logic control synthesis with probabilis- tic satisfaction guarantees. In 2019 IEEE 58th Conference on Decision and Control (CDC), pp. 5338–5343. IEEE, 2019.

Nils Jansen, Bettina Könighofer, Sebastian Junges, Alex Serban, Roderick Bloem:
Safe Reinforcement Learning Using Probabilistic Shields. CONCUR 2020: 3:1-3:16


**Strength And Weaknesses:**

+ The paper is easy to understand
+ The key concept is clear and important, aiming to bringing learning under/with/via temporal constraints differentiable and thereby more scaling
+ The experimental evaluation is extensive and convincing
- The connection to probabilistic temporal logic constraints is unclear
- At parts, the paper could be better polished (mostly in terms of grammar)


**Summary Of The Paper:**

This paper concerns hierarchical reinforcement learning (RL) in safety critical scenarios. The critical aspect is captured via the inclusion of temporal logic constraints, in fact (a variant of) LTL. The key feature of this paper is that a quantitative notion of LTL is defined and used, to get a differentiable logic as opposed to the more binary nature of standard LTL (at a state: formula is satisfied or not). The RL procedure is defined in a hierarchical way, where high-level planning is complemented by low-level execution of a controller. The authors provide all necessary definitions, explain their variant of a quantitative notion, and provide an extensive evaluation of their approach on rather challenging environments.

**Summary Of The Review:**

Good paper, related work/novelty can be better argued.

---

> ### Author Response · Authors · 2022-11-19
> **Response to Reviewer KU3V**
>
> We would like to express our deepest gratitude for your constructive feedback again! We have revised our manuscript in response to discuss the probabilistic temporal logic constraints and shield in related work.
>
> ### Probabilistic Temporal Logic Constraints
> [1,2] propose techniques to synthesize controllers or shields with probabilistic guarantees for satisfying LTL specifications under uncertainty. The uncertainty is embedded in the environment (e.g., the uncertainty in the structure of the workspace, or the agent's actions). Such uncertainty also exists in our environments manifesting as stochastic dynamics and policies. Modeling these probabilistic behaviors formally and providing probabilistic guarantees of satisfying LTL specifications in hierarchical settings is an interesting direction for future work.
>
> ### MDP Semantics, Safe RL, and Their Differentiability
> For most practical problems, the MDP model is usually unknown (especially in model-free deep reinforcement learning). Even if they are known and differentiable, there is still a gap in leveraging this differentiability for LTL controller/policy synthesis. We do not realize there exists techniques can directly differentiate through the MDP and constrained policy update like our approach.
>
> Existing safe RL work, such as CPO [3] is also based on the Lagrangian method. The constraint functions in these works are indeed differentiable. However, they do not explicitly model formal constraints as we do. Instead, they only learn neural network surrogate constraint functions. Unlike a differentiable STL constraint function, a neural network surrogate function cannot be precisely customized to take users' domain knowledge into account (e.g., in a loop task, users may want a drone to hover around the duck without colliding with it).
>
> ### Misc.
> * We shrank the introduction to LTL, and left a brief description for readers who might not be familiar with the topic.
> * We included more related content about probabilistic LTL constraints and reward shaping in the related work section.
>
> **Reference**
> [1] Mohammadhosein Hasanbeig, Yiannis Kantaros, Alessandro Abate, Daniel Kroening, George J Pappas, and Insup Lee. Reinforcement learning for temporal logic control synthesis with probabilistic satisfaction guarantees. In 2019 IEEE 58th Conference on Decision and Control (CDC), pp. 5338–5343. IEEE, 2019.
>
> [2] Nils Jansen, Bettina Könighofer, Sebastian Junges, Alex Serban, Roderick Bloem: Safe Reinforcement Learning Using Probabilistic Shields. CONCUR 2020: 3:1-3:16
>
> [3] Achiam, Joshua, et al. "Constrained policy optimization." International conference on machine learning. PMLR, 2017.

---

> > ### Comment · Reviewer_KU3V · 2022-12-03
> > **Thank you..**
> >
> > ..for the clarification and updated version of the paper. I tend to keep my original score of weak accept.

---

### Official Review · Reviewer_urxX · 2022-11-01

**Confidence:** 4
**Correctness:** 4
**Technical Novelty And Significance:** 3
**Empirical Novelty And Significance:** 3
**Recommendation:** 8

**Clarity, Quality, Novelty And Reproducibility:**

The paper is written rather well in terms of flow, but I feel also that it undersells the idea. There are a few nits here and there, such as the fact that the term "SDF function" parses out to "signed distance function function". Nothing terrible, but a careful read-over would be beneficial. Regarding quality and novelty, my general feelings as expressed in the strengths/weaknesses field are that this is good quality work, presented suboptimally. The suboptimality is mostly optical though, the actual exposition is quite clear and the ideas are easily understood. The relative low novelty is offset by the elegance with which the elements of the framework come together.

**Strength And Weaknesses:**

Weaknesses:
1. The general novelty of the approach is not that high, as it combines building blocks that are well understood (there is more to be said on this in strengths section).
2. This paper is written in a way that is honestly kind of boring and doesn't sell the approach particularly well. 70% of this paper is background and everything interesting is kind of compressed at the end in tables or in the appendix. In some sense, it feels like the goal of this paper is to push benchmarks, not to convince the reader that they could be solving new and interesting problems that can be made more tractable by injecting domain knowledge in the form of constraints. If I were to rewrite this paper, I would put the highlight on concrete problems that are intractable using standard DRL, for example, a focus could be an illustration of how a formal specification provides a principled way out of the nightmare that is reward shaping, in particular with a focus on the additional constructs enabled by STL.

Strengths:
1. I think the problem tackled here is really important and generally underappreciated (see above).
2. While none of the building blocks are particularly novel, it has to be noted that they come together rather well in this paper. The proposed solution is rather elegant without any obviously awkward bits and I could see this as an actual tool that one would productively use for tackling a problem where the policy lends itself to an STL specification.

**Summary Of The Paper:**

This paper presents an approach to constrained deep reinforcement learning using signal temporal logic as a constraint language. The formulas defining the constraints are converted to algebraic constraints using the quantitative semantics of STL, which, in turn are incorporated as Lagrange terms in a dual ascent algorithm. These concepts are explored within the space of hierarchical RL. The results are evaluated on benchmark domains.

**Summary Of The Review:**

I think that this paper is likely to be of interest to a non-trivial number of people. The work is technically reasonable and the ideas are quite nice.

---

> ### Author Response · Authors · 2022-11-19
> **Response to Reviewer urxX**
>
> Thanks for your positive review of our paper!
>
> Indeed, we wish to highlight that our solution is a more tractable technique than existing approaches. To further buttress this claim, we have added the following:
>
> 1. We illustrate sample complexity issues of LTL reward shaping in the first paragraph of the introduction. The main argument here is that the policy gradient estimated from samples can provide a poorer signal than the gradient directly computed from the STL specification.
> 2. Unlike learning a single goal-conditioned policy, we find the existing hierarchical approaches with temporal logic instructions that learn multiple option policies to scale poorly with specification complexity as now mentioned in the third paragraph of the introduction.
>
> These two claims are supported by comparison to the existing reward shaping work in Fig.3 and Fig.12.  Additionally, we now provide two more ablation studies (Table.2 and Table.6 in the revision) to further justify these claims.  The code for these experiments is also available in the anonymous repository provided in the replication statement.
>
> Our revision provides additional elaboration on these points in the introduction's contribution summary.

---

### Author Response · Authors · 2022-11-19
**Many thanks to all the reviewers!**

We thank reviewer urxX for the positive review and constructive suggestions on the revision. We thank reviewer KU3V for providing related work on probabilistic LTL constraints and shields. We thank reviewer Ea5U for bringing the STL-related work into context. We thank reviewer Pa7M's insights on STL-related work and for pointing out several valuable papers in this field.

All the changes we made in the revision have been highlighted with magenta. The code for all experiments generated for the rebuttal is also available in the anonymous repository in the replication statement.

---

### Decision · Program_Chairs · 2023-01-20

**Decision:**

Reject

**Justification For Why Not Higher Score:**

The weaknesses listed above seem serious enough that this is not a viable paper.

**Justification For Why Not Lower Score:**

This is the lowest score.

**Metareview: Summary, Strengths And Weaknesses:**

Strengths:  The paper is a nice integration of existing ideas together with a coherent specification.

Weaknesses:  This is an integration paper rather than a paper with significantly new ideas.  While the integration is nice, the deep alternation of min-max quantifiers undermines scalability.  As an integration paper the experimental results are important.  The experimental results seem weak --- the ablation studies fail to confirm the value of some of the central components.

**Summary Of Ac-Reviewer Meeting:**

The initial scores ranged from 3 to 8.  In the zoom meeting the reviewer giving a 3 made a series of very coherent and well thought out points about weaknesses of the paper described in the weakness section above.  The advocate for the paper did not present a compelling case and agreed to reduce their score.